# Interconnected subsets of memory follicular helper T cells have different effector functions

Assia Asrir[1,2,3,4], Meryem Aloulou[1,2,3,4], Mylène Gador[1,2,3,4], Corine Pérals[1,2,3,4] & Nicolas Fazilleau[1,2,3,4]

Follicular helper T cells regulate high-affinity antibody production. Memory follicular helper T cells can be local in draining lymphoid organs and circulate in the blood, but the underlying mechanisms of this subdivision are unresolved. Here we show that both memory follicular helper T subsets sustain B-cell responses after reactivation. Local cells promote more plasma cell differentiation, whereas circulating cells promote more secondary germinal centers. In parallel, local memory B cells are homogeneous and programmed to become plasma cells, whereas circulating memory B cells are able to rediversify. Local memory follicular helper T cells have higher affinity T-cell receptors, which correlates with expression of peptide MHC-II at the surface of local memory B cells only. Blocking T-cell receptor–peptide MHC-II interactions induces the release of local memory follicular helper T cells in the circulating compartment. Our studies show that memory follicular helper T localization is highly intertwined with memory B cells, a finding that has important implications for vaccine design.

[1] Centre de Physiopathologie de Toulouse Purpan, Toulouse F-31300, France. [2] INSERM, U1043, BP 3028, 31024 Cedex 3, Toulouse F-31300, France. [3] CNRS, UMR5282, Toulouse F-31300, France. [4] Université Toulouse III Paul-Sabatier, Toulouse F-31300, France. Correspondence and requests for materials should be addressed to N.F. (email: nicolas.fazilleau@inserm.fr)

Most effective vaccines in use rely on the long-term protection of high-affinity memory B cells and long-lived plasma cells. Particularly, B-cell responses to protein antigens (Ag) develop under the guidance of follicular helper T (Tfh) cells. Effector Tfh cells develop locally in lymphoid organs draining the site of immunization[1]. These cells regulate the outcome of humoral responses through a combination of specific T-cell receptor (TCR) interactions with peptide-MHCII (pMHCII), engagement of co-stimulatory molecules and cytokine delivery[2, 3]. These events result in class-switch recombination and somatic diversification of the B-cell receptor (BCR) in the germinal center (GC) and, ultimately, the selection of high-affinity B-cell variants into the plasma cell and memory B-cell compartment. The transcriptional regulator Bcl-6 drives the differentiation of this specific helper T (Th) cell lineage[4]. Bcl-6 induces the expression of the chemokine receptor CXCR5, a hallmark of Tfh cells, which promotes their migration in

CXCL13-rich areas such as B follicles. Furthermore, ICOS-ICOS-L engagement induces differentiation and maintenance of Tfh cells and ICOS expression by Tfh cells is mandatory for GC formation[5]. Another distinguishing feature of Tfh cells is the expression of programmed cell death-1 (PD-1), an inhibitory receptor expressed highly by GC Tfh cells[6]. Finally, effector Tfh cells produce large amounts of IL-21, the most potent cytokine known to drive plasma cell differentiation[7, 8] and optimal Bcl-6 expression in GC B cells[9, 10]. Effector Tfh cells can also secrete other cytokines, such as IL-4[11], IL-17,[12] or IFN-γ[13] that, in this context, control class-switch recombination.

Until recently, Tfh cells were considered as fully differentiated effector cells prone to apoptosis while the GC reaction resolved[14, 15]. However, we detected memory CXCR5+ Th cells after protein vaccination in draining lymphoid tissue[1]. The existence of memory Tfh cells has now been demonstrated in both mice[16–18] and humans[19–22]. By using cell transfer experiments, Liu et al.[23]

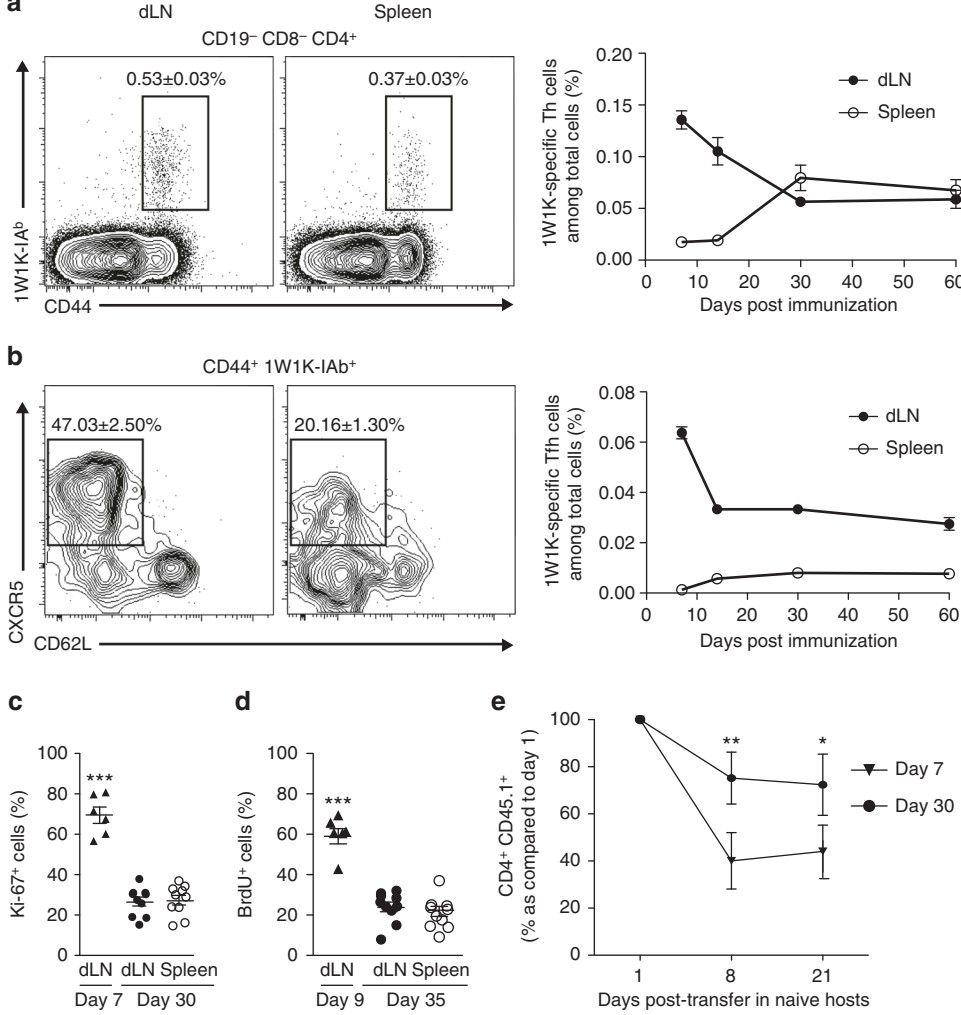

**Fig. 1** Tracking local and circulating 1W1K-specific memory Tfh cells. Thirty days after sc immunization with 1W1K-OVA, dLN and spleen were analyzed for the detection of 1W1K-specific activated CD4+ T cells (1W1K-IAb+CD44+) (**a**) and 1W1K-specific Tfh cells (CXCR5+CD62L−) (**b**). Percentages among CD4+ T cells and among 1W1K-IAb+CD44+ are depicted (mean ± SEM, n ≥ 22). Kinetics of 1W1K-specific Th cells and 1W1K-specifc Tfh cells are shown (percentages among total live cells are depicted, mean ± SEM, n = 5/time point). Intracellular expression of Ki-67 among 1W1K-specific Tfh cells at day 7 and day 30 post immunization (**c**). Day 27 or day 1 post immunization, mice were injected ip with BrdU or PBS every 48 h for 1 week. Frequency of BrdU+ among 1W1K-specific Tfh cells (**d**). C57BL/6 mice were transferred iv with 100,000 purified CD4+CD45.1+ naive OT-II cells and sc immunized 24 h later. After 7 or 30 days, dLN CD4+CD45.1+CD44+CXCR5+OT-II cells were purified and transferred iv in naive hosts. Days 1, 8, and 21 post transfer, mice were killed and spleen cells analyzed (**e**) (percentages of transferred cells as referred to day 1). Each *dot* represents an individual mouse; *horizontal lines* denote the mean value of groups and SEM. Data are representative of at least three independent experiments. Mann–Whitney test, *P < 0.05; **P < 0.01; ***P < 0.001

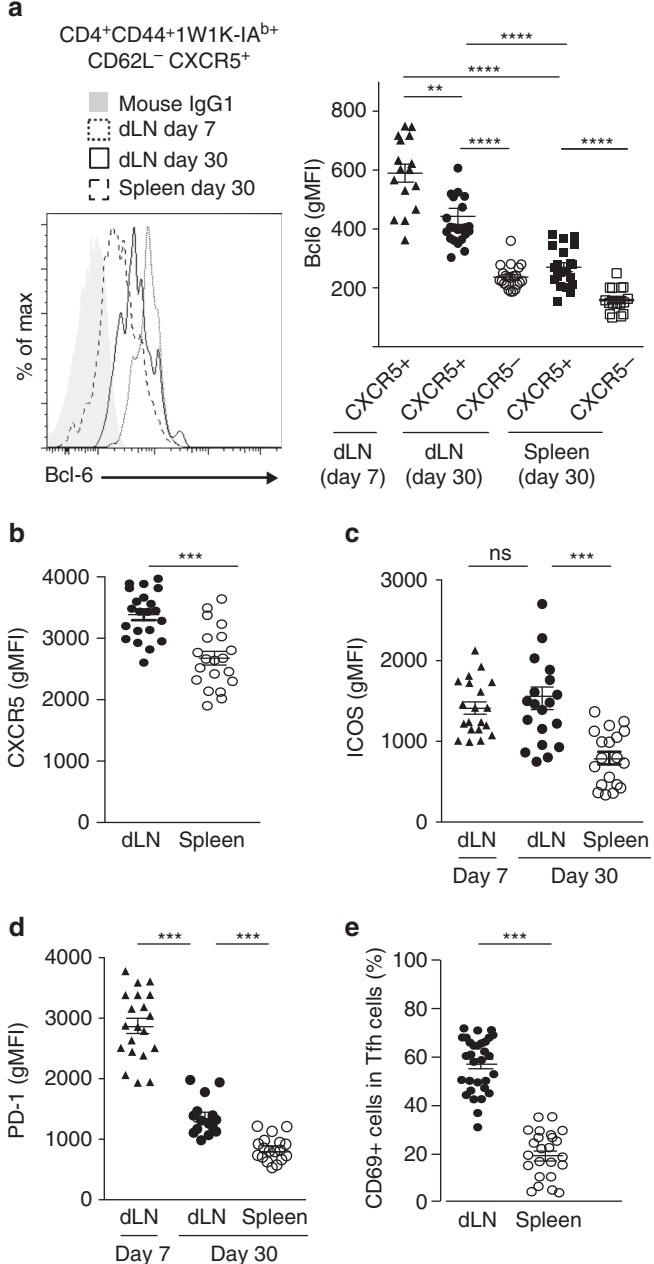

**Fig. 2** Phenotype of local and circulating 1W1K-specific memory Tfh cells. Expression level of Bcl-6, ICOS, and PD-1 of 1W1K-specific memory Tfh cells as compared to 1W1K-specific effector Tfh cells (**a**, **c**, **d**). Expression level of CXCR5 (**b**) and frequency of CD69+ (**e**) cells among 1W1K-specific memory Tfh cells. Each *dot* represents an individual mouse; *horizontal lines* denote the mean value of groups and SEM. Data are representative of five independent experiments. Mann–Whitney test; ns, non-significant; **$P < 0.01$; ***$P < 0.001$; ****$P < 0.0001$

demonstrated that memory Bcl-6+CXCR5+ Th cells are the most likely cells to become effector Tfh cells upon reactivation, thus defining memory Tfh cells. The latter are resting cells that can be long-lived[18]. The differentiation of these cells is still not totally understood, but differentiation of a memory Tfh cell does not seem to require participation in the GC response[24]. Interestingly, Bcl-6 expression in memory Tfh cells is decreased as compared to with effector Tfh cells[23, 25, 26]. Consequently, memory Tfh cells are committed to the Tfh lineage, but with a less polarized phenotype than their effector counterparts[18, 27, 28].

One important attribute of memory Tfh cells is their localization. We have previously shown that memory Tfh cells are present predominantly in draining lymph nodes (dLNs) where they form a local pool[1]. This localization probably results as retention of memory Th cells in dLN correlates with a prolonged exposure of Ag[29], that persistent Ag is crucial to sustain the Tfh phenotype[30], and that depots of pMHCII persist in the dLN after immunization[1], even if the nature of the Ag-presenting cells in the memory phase is unknown. By contrast, circulating memory Tfh cells can also be detected in the blood of mice[18] and humans[19, 20, 31]. Similarly, multiple subsets of memory B cells exist and colonize different localizations, for example the long-lived plasma cells niche in the bone marrow and memory B cells circulate in second lymphoid organs. In addition, the existence of two Ag-specific memory B-cell subsets with distinct functional capacities have been described[32–35]. Upon Ag recall, some memory B cells enter the GC to rediversify the BCR, while other memory B cells differentiate into Ab-secreting plasma cells. Interestingly, after Ag reactivation, memory B cells induce rapid effector functions of memory Tfh cells, establishing the close relationship between memory B cells and memory Tfh cells[36].

Although the phenotype and function of memory B cells are well described, whether local and circulating memory Tfh cells are phenotypically and functionally distinct, as well as whether these cell subsets are inter-connected, is unknown. Here we use different Ag models with which we can track Ag-specific T and B cells in C57BL/6 mice after protein immunization. We show that local memory Tfh cells have a more Tfh-polarized phenotype than the circulating memory Tfh cells. Local memory Tfh cells express high-affinity TCRs and localize preferentially in B follicles in the proximity of local memory B cells. Notably, both memory Tfh cell subsets promote B-cell responses after Ag re-challenge. Local memory Tfh cells promote plasma cell differentiation, whereas circulating memory Tfh cells are more prone to redirect B cells into the GCs. In parallel, local memory B cells form a homogeneous population programmed to become plasma cells after Ag recall, whereas circulating memory B cells form a heterogeneous population with cells capable of participating to secondary GCs. Interestingly, long-term pMHCII expression is observed at the surface of local memory B cells only. In vivo blocking of TCR–pMHCII interactions induces the release of local memory Tfh cells into the circulating compartment. Thus, this study reveals that the anatomical localization of memory Tfh cell is highly intertwined with memory B cells, a finding that may have important implications for vaccine design.

## Results

**Local memory Tfh cells have a more Tfh-polarized phenotype.**
Subcutaneous (sc) immunization with the protein formed of the peptide variant (EAWGALANKAVDKA, called hereafter 1W1K) of the I-E alpha chain immunodominant peptide 52–68 (Eα52-68) conjugated to ovalbumin (OVA) in incomplete Freund's adjuvant (IFA) together with the TLR9 agonist CpG favors robust Tfh-dependent B-cell immunity[37]. 1W1K-I-A$^b$ pMHCII tetramer staining together with cell surface expression of CD44 provided direct access to Ag-specific Th cells from sc immunized C57BL/6 mice in both dLN and the non-draining spleen, the organ that filters the blood in which we can track circulating cells (Supplementary Fig. 1 and Fig. 1a). While Ag-driven cell expansion reached peak levels by day 7 post immunization in the dLN, there was a rapid contraction in the next 3 weeks followed by a stabilization phase that typically followed the resolution of the primary response (Fig. 1a). In the spleen, the kinetics of the Th cell response was different with a higher proportion of 1W1K-specific Th cells at day 30 that stabilized over time. To assess the kinetics

of the Tfh response, we analyzed the 1W1K-specific Th cells that were CD62L$^{lo}$CXCR5$^+$ as a broad indicator of activated Th cells relocating to the T/B border[3]. We found CXCR5$^+$ Th cells only in the dLN but not in the spleen at day 7 after immunization (Fig. 1b). However, as previously described in the blood[24], CXCR5$^+$ Th cells circulated in the spleen of C57BL/6 mice from

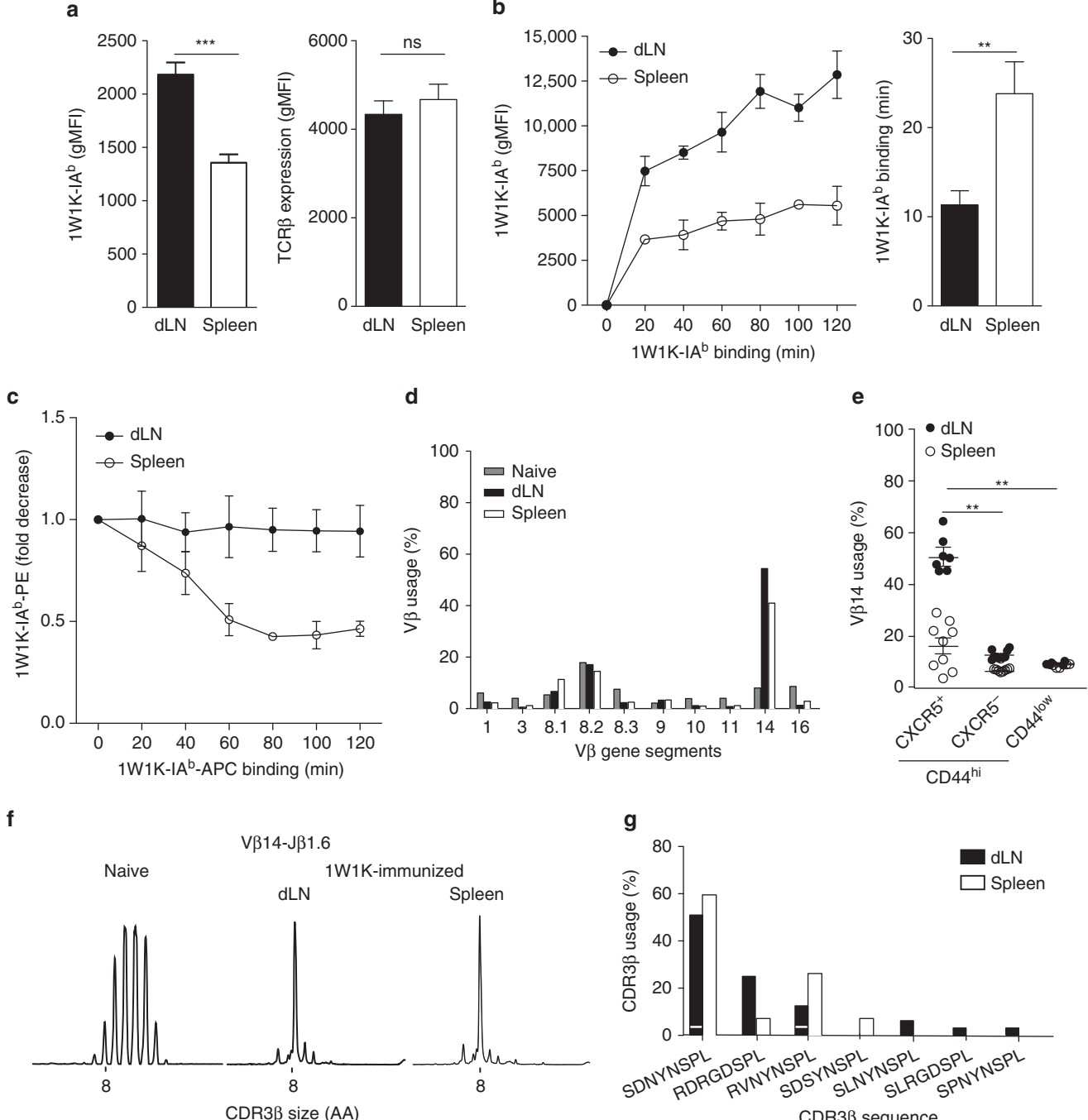

**Fig. 3** Local 1W1K-specific memory Tfh cells bear high-affinity TCR. gMFI of 1W1K-IA$^b$ tetramer and TCRβ staining at the surface of dLN and spleen 1W1K-specific mTfh cells (1W1K-IA$^b$+CD44$^+$CXCR5$^+$CD62L$^-$) at day 30 post immunization (**a**; mean ± SEM, $n \geq 22$). gMFI of 1W1K-IA$^b$ tetramer staining at the surface of day 30 1W1K-specific memory Tfh cells after staining with varying times of pMHCII tetramer (**b**, *left*; mean ± SEM, $n \geq 4$/time point). Estimated mean of time of pMHCII tetramer staining in minutes for each subset to reach a gMFI of 3930 is also depicted (**b**, *right*; mean ± SEM, $n \geq 4$). Decay is represented in fold decrease (the reference corresponds to gMFI of PE-1W1K-IA$^b$ tetramer staining at the surface of day 30 1W1K-specific mTfh (from dLN or spleen) after staining with optimal concentration and time of PE-MHCII PE-1W1K-IA$^b$ tetramer and after addition with varying times of APC-pMHCII tetramer (**c**, mean ± SEM, $n = 5$). Vβ usage by quantitative RT-PCR from naive Th cells or day 30 dLN and spleen 1W1K-specific memory Tfh cells purified and pooled from 20 immunized mice (**d**). Frequency of Vβ14 by flow cytometry among 1W1K-specific CD44$^{hi}$ CXCR5$^+$ or CXCR5$^-$ as compared to CD44$^{low}$CD4$^+$ T cells. Each *dot* represents an individual mouse; *horizontal lines* denote the mean value of groups and SEM (**e**). CDR3β length distribution for Vβ14-Jβ1.6 rearrangement from naive CD4$^+$ T cells or day 30 dLN and spleen 1W1K-specific memory Tfh cells purified and pooled from 20 immunized mice (**f**). Usage of the different CDR3β aa sequences. A *white stripe* on the *black bar* separates the different nucleotidic sequences among unique aa sequence (**g**). Data are representative of at least three independent experiments. Mann–Whitney test; ns, non-significant; **$P < 0.01$; ***$P < 0.001$

day 10 post immunization (Fig. 1b). Strikingly, by day 30 post immunization, two stabilized populations of CXCR5[+] Th cells were found local in the dLN and circulating in the spleen (Fig. 1b). While around 70% of effector Tfh cells in the dLN 7 days after immunization were Ki-67[+], only 20% of the CXCR5[+] Th cells in the dLN and in the spleen were at day 30 post immunization (Fig. 1c). These results were confirmed by BrdU staining (Fig. 1d), thus demonstrating that dLN and spleen CXCR5[+] Th cells were non-proliferating in contrast to their effector counterparts. Notably, after adoptive transfer of naive TCR transgenic Th cells in C57BL/6 recipients followed by sc immunization, we also observed a division in localization for the transgenic CXCR5[+] Th cells in the spleen and in the dLN 30 days after immunization (Supplementary Fig. 2). Moreover, when TCR transgenic CXCR5[+] Th cells from day 30 sc immunized mice were purified and transferred into naive hosts, they were still present 21 days post transfer while the majority of the transferred CXCR5[+] Th cells from day 7 sc immunized mice disappeared (Fig. 1e). Thus, CXCR5[+] Th cells at day 30 post immunization can be classified as memory cells. Overall, sc immunization allows the development of memory Tfh cells, with some cells remaining local in the dLN and some cells circulating in the spleen.

How these two memory CXCR5[+] Th cell subsets relate to specialized Tfh cells was not clear since their detection was based on CD62L downregulation and CXCR5 expression. Therefore, we monitored intra-cellular expression of Bcl-6 as well as surface expression of molecules expressed by effector Tfh cells such as ICOS, PD-1, and CD69[2, 4]. We found that dLN and spleen memory CXCR5[+] Th cells expressed lower levels of Bcl-6 than day 7 effector Tfh cells, with a more pronounced decrease for spleen memory cells (Fig. 2a). However, expression levels of Bcl-6 were significantly higher in these memory CXCR5[+] Th cells than in their CXCR5[−] Th cell counterparts (Fig. 2a). Further, dLN memory Tfh cells expressed higher level of CXCR5 than spleen memory Tfh cells (Fig. 2b). In addition, Tfh cells in the dLN expressed higher level of ICOS than spleen memory Tfh cells, irrespective of whether they were effector or memory cells (Fig. 2c). Strikingly, as observed for Bcl-6, both memory Tfh cells expressed PD-1 to a lower level than effector Tfh cells, but dLN memory Tfh cell expression was higher than spleen memory Tfh cell one (Fig. 2d). Interestingly, these higher CXCR5 and PD-1 expression levels were also demonstrated for local memory Tfh cells in the spleen following intraperitoneal (ip) injection as compared to circulating memory Tfh cells in the spleen after sc immunization (Supplementary Fig. 3). Thus, the difference between dLN and spleen memory Tfh cells after sc immunization was not due to the lymphoid tissue itself but rather to the site of Ag entry. Finally, we monitored CD69 expression at the surface of the memory Tfh cells subsets. CD69 is rapidly upregulated after activation and late CD69 expression can be seen at the surface of effector Tfh cells[38]. Moreover, CD69 expression distinguishes tissue-resident memory T cells from circulating memory T cells in humans[39]. Here we found that CD69 expression also persisted in the memory phase on a majority of memory Tfh cells in the dLN (Fig. 2e). In contrast, almost no memory Tfh cells in the spleen were CD69[+]. Overall, local memory Tfh cells are Bcl-6[+]CXCR5[+]ICOS[hi]PD-1[+]CD69[+] while circulating memory Tfh cells are Bcl-6[lo]CXCR5[lo]ICOS[+]PD-1[lo]CD69[−]. Notably, the memory Tfh cell subdivision was also observed even 120 days post immunization with a large proportion of local memory Tfh cells still expressing CD69 at this late time point (Supplementary Fig. 4) or using non-depot adjuvants such as squalene-based adjuvant or aluminum salts (Supplementary Fig. 5). Moreover, this memory subdivision was also observed in the dLN and the spleen after intranasal infection using PR8 strain of influenza (Supplementary Fig. 6). Thus, over a Tfh-dependent B-cell

responses, memory Tfh cells persist long term in vivo. These memory Tfh cells can be subdivided in local memory cells in the draining lymphoid organ with a more Tfh-polarized phenotype than their circulating memory Tfh cell counterparts.

**Local memory Tfh cells are preferentially of high affinity**. To better decipher the subdivision of the memory Tfh cells, we focused on the local and circulating memory CXCR5[+] Th cells, respectively, in the dLN and the spleen after sc immunization. We first analyzed the TCR affinity of these cells. We found that spleen memory Tfh cells had a lower binding to pMHCII tetramer than their dLN counterparts (Fig. 3a). This difference was not due to a decrease in the TCR expression level as measured by the TCRβ gMFI (Fig. 3a). Strikingly, this observation was also found at the surface of local memory Tfh cells after ip immunization in the draining spleen (Supplementary Fig. 2C) and in the draining mediastinal LN after intranasal influenza infection (Supplementary Fig. 5E). Thus, the TCR borne by local memory Tfh cells have greater capacities to bind their cognate ligand than the circulating memory Tfh cells.

We then quantified the dynamics of pMHCII tetramer staining as an assessment of TCR avidity. We first estimated the association of pMHCII tetramer to TCR at the surface of the different memory Tfh cells. Using optimal concentration of pMHCII tetramer, we varied the time of staining. While the pMHCII staining at the surface of spleen memory Tfh cells reached a gMFI superior or equal to 3930 after a labeling time of $23.8 \pm 3.5$ min, dLN memory Tfh cells reached this same level with a statistically lower time ($11.33 \pm 1.5$ min; Mann–Whitney test, $P < 0.01$, Fig. 3b). Further, we estimated the dissociation of pMHCII tetramer complexed with TCR at the surface of the different memory Tfh cell subsets (Fig. 3c). Cells were stained as usual with PE-labeled pMHCII tetramer and put in presence of APC-labeled pMHCII tetramer as a competitor for TCR complexing. No decay of the PE-labeled pMHCII labeling was observed for dLN memory Tfh cells while the decay was around 50%, thus greater, for the spleen memory Tfh cells (Fig. 3c). Hence, TCR used by local mTfh cells have longer binding to pMHCII.

To assess the potential differences in TCR diversity, we studied the TCR repertoire of 1W1K-specific memory Tfh cells at the molecular level. We found that the Vβ14 gene segment was highly represented in both dLN and spleen memory Tfh cells at the message level by quantitative RT-PCR (Fig. 3d) and at the protein level by flow cytometry (Fig. 3e). Strikingly, the Vβ14 gene segment was more dominant in dLN than in spleen memory Tfh cells ($51.56 \pm 2.60\%$ and $16.38 \pm 3.04\%$, respectively; Mann–Whitney test, $P < 0.01$, Fig. 3e). T cell repertoire studies have previously shown that T cells specific for one distinct Ag express TCR with Vβ-Jβ rearrangement common to all mice of the same MHC haplotype[40]. These rearrangements are named public. In order to identify it in the context of the Th cell response to 1W1K, CDR3β length distribution for the Vβ14 gene segment was estimated. While a bell-shaped curve characteristic of polyclonal repertoire was found in Th cells from a naive mouse, an expansion of the Vβ14-Jβ1.6 rearrangement corresponding to a CDR3 of 8 amino acids (aa) was observed in both dLN and spleen memory Tfh cells (Fig. 3f). CDR3β sequencing was performed and showed that the majority of the clonal expansion corresponded to three different aa sequences (SDNYNSPL, RDRGDSPL, and RVNYNSPL; Fig. 3g). It is worth noticing that these aa sequences were encoded by different nucleotidic sequences, indicating the assortment for these rearrangements during the clonal selection. Other CDR3β sequences were also found but to a lower frequency and only one in the local memory Tfh cell subset.

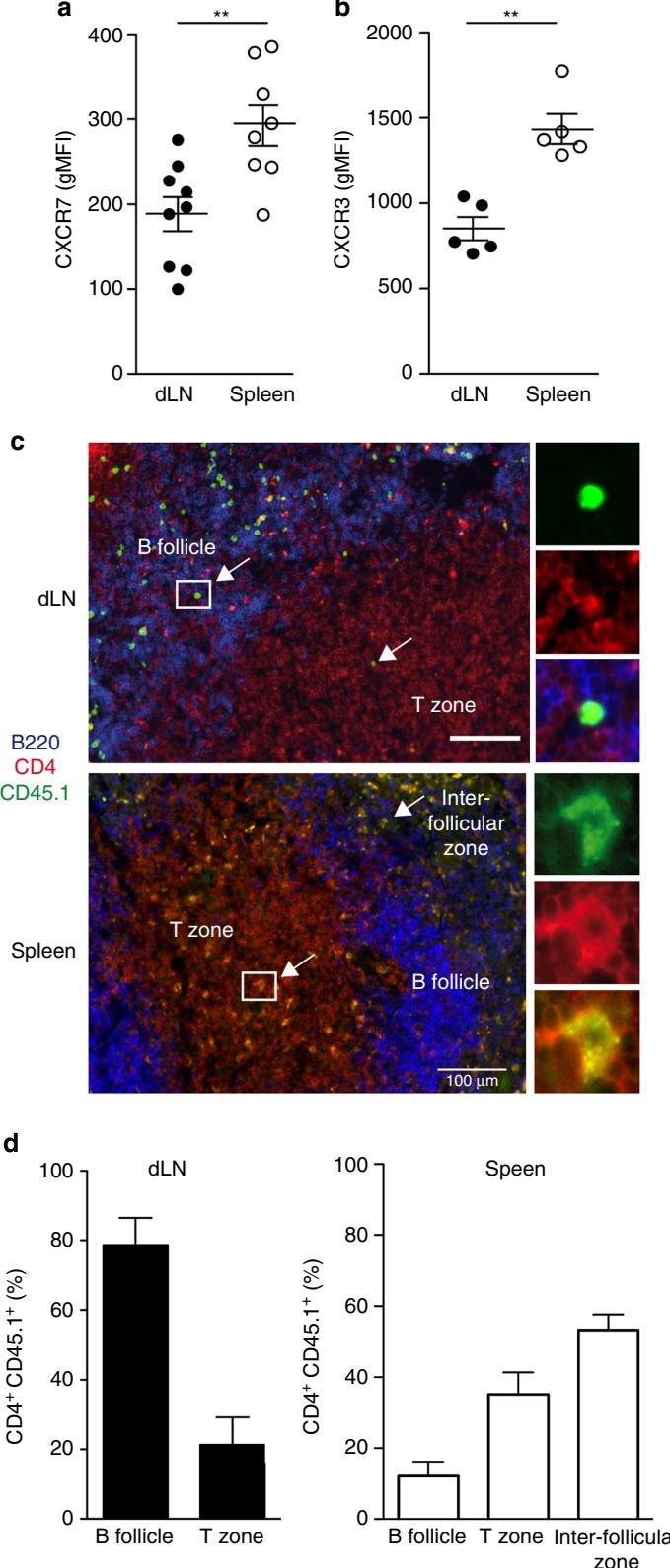

**Fig. 4** Local 1W1K-specific memory Tfh cells localize preferentially in the B-cell follicle. Thirty days after sc immunization, dLN and spleen 1W1K-specific memory Tfh cells (1W1K-IA$^{b+}$CD44$^+$CXCR5$^+$CD62L$^-$) were analyzed for CCR7 (**a**) and CXCR3 (**b**). Each *dot* represents an individual mouse; *horizontal lines* denote the mean value of groups and SEM. A number of 10$^6$ naive CD4$^+$CD45.1$^+$ OT-II cells were injected iv into C57BL/6 mice that were sc immunized the day after. After 30 days, confocal microscopy studies of dLN and spleen were performed using anti-B220 (*blue*), anti-CD4 (*red*), and anti-CD45.1 (*green*) mAb. Localization of CD4$^+$CD45.1$^+$ cells in B follicle (B220$^+$), T zone (CD4$^+$), and inter-follicular zone is highlighted (**c**; *scale bar*, 100 μm). CD4$^+$CD45.1$^+$ cell distribution in the different areas was quantified using IMARIS (**d**). Data are representative of at least three independent experiments. Mann–Whitney test, **$P < 0.01$

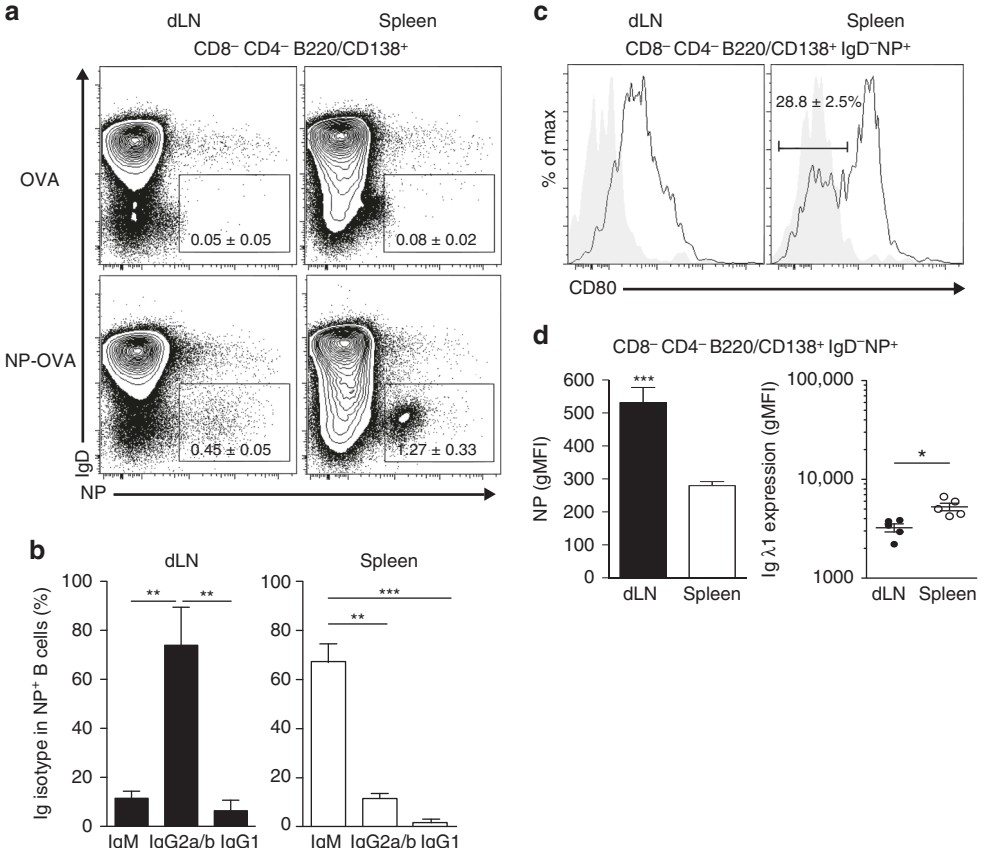

**Fig. 5** Tracking local and circulating NP-specific memory B cells. C57BL/6 mice were immunized sc with 100 μg OVA or NP-OVA. *Dot plots* of NP-specific B cells at day 30 post immunization in the dLN and in the spleen (see Supplementary Fig. 7 for gating strategy; **a**; mean ± SEM, $n = 5$). Ig isotype distribution among IgD⁻NP⁺ B cells at day 30 post immunization (**b**; mean ± SEM, $n \geq 4$). CD80 expression at the surface of dLN and spleen IgD⁻NP⁺ B cells (**c**; mean ± SEM, $n = 5$). Histograms in *gray* represent isotype controls. gMFI of NP and Igλ1 staining at the surface of IgD⁻NP⁺ B cells at day 30 post immunization (**d**; mean ± SEM, $n = 5$). Data are representative of at least three independent experiments. Mann–Whitney test, $*P < 0.05$; $**P < 0.01$; $***P < 0.001$

Overall, the dominant and public TCR repertoire borne by memory Tfh cells is similar between local and circulating cells. Anyhow, the public TCR repertoire is more abundant in the local pool, which could result in higher avidity for the pMHCII of the local memory Tfh cells as compared to the circulating memory Tfh cells.

**Local not circulating memory Tfh cells localize in B follicles**. We next assessed chemokine receptor expression at the surface of the memory Tfh cells. As described above, we found that 1W1K-specific memory Tfh cells in the dLN expressed higher levels of CXCR5 than spleen memory Tfh cells (Fig. 2c). In contrast, we found that spleen memory Tfh cells expressed higher levels of CCR7, the receptor of CCL19 and CCL21 (Fig. 4a). We also monitored the expression of CXCR3 and CCR6, two chemokine receptors expressed by Tfh cells that generally associate with human Th1 and Th17, respectively[20]. After sc immunization with IFA + CpG, a type 1 immunity adjuvant, we found that neither dLN nor spleen memory Tfh cells expressed CCR6. In contrast, as expected after IFA + CpG immunization, we found that both memory Tfh cell subsets expressed CXCR3 but with lower level for local dLN cells than for circulating spleen cells (Fig. 4b). Using OVA-specific TCR transgenic OT-II cell transfer, we detected by confocal microscopy 30 days after sc immunization the localization of local and circulating memory OT-II cells in the dLN and in the spleen respectively (Fig. 4c). CD4⁺CD45.1⁺OT-II cell distribution in the different areas was quantified and demonstrated that memory OT-II cells were in great majority in the B follicle in

the dLN and outside the B follicle in the spleen (Fig. 4d). Altogether, our data suggest that local memory Tfh cells express chemokine receptors that allow their localization in B follicle while circulating memory Tfh cells remain mainly outside of the B follicle.

**Local memory B cells are preferentially of high affinity**. The interdependency between Tfh and B cells in the effector phase has been described, but this phenomenon in the memory phase remains unappreciated. We took advantage of the 4-hydroxy-3-nitrophenylacetyl (NP)-OVA-conjugated protein with which we can track NP-specific B cells from immunized C57BL/6 mice by flow cytometry[37]. We could detect NP-specific CD138/B220⁺IgD⁻ cells in the dLN and in the spleen at day 30 after sc immunization as compared to mice sc immunized with OVA (Supplementary Fig. 7 and Fig. 5a). These cells were memory cells since they were CD38⁺ (Supplementary Fig. 8A). Moreover, looking at the kinetics of the NP-specific B-cell response, we found GL-7⁺CD38⁻ GC B cells in the dLN but not in the spleen thus the GC reaction occurred only locally in the dLN after sc immunization (Supplementary Fig. 8B). Strikingly, the GC reaction was ended by day 30 post immunization since almost no NP-specific GL-7⁺CD38⁻ GC B cells were detected at this later time point by flow cytometry (Supplementary Fig. 8B) or by immunofluorescence (Supplementary Fig. 9). Recent studies pointed to the existence of two dedicated memory B-cell subsets with distinct functional capacities[32–35]. After re-exposure to Ag, the first cell subset enters predominantly in GC while the second

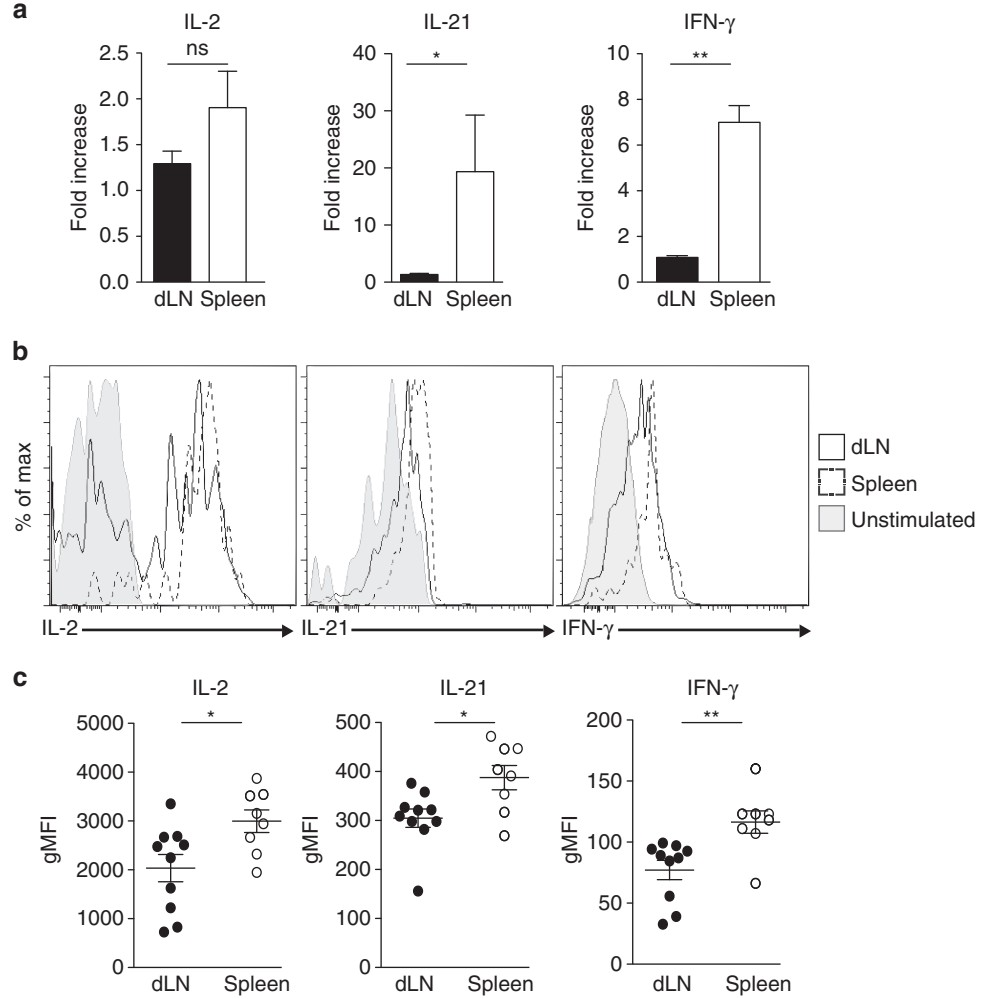

**Fig. 6** Circulating memory Tfh cells produced more cytokines after restimulation. C57BL/6 mice were transferred iv with $10^5$ purified CD4$^+$CD45.1$^+$ naive OT-II cells and sc immunized 24 h later with OVA in IFA/CpG. After 30 days, dLN and spleen CD4$^+$CD45.1$^+$CD44$^+$CXCR5$^+$OT-II cells were purified and stimulated in vitro with anti-CD3/CD28 beads. After 48 h, cells were collected and their RNA was extracted and retro-transcribed into cDNA. qPCR was then performed to evaluate the transcript level (**a**; mean ± SEM, n = 4). In parallel, dLN and spleen CD4$^+$CD45.1$^+$CD44$^+$CXCR5$^+$OT-II cells were stimulated in vitro with PMA/ionomycin and intracellular staining for IL-2, IL-21, and IFN-γ performed (**b**, **c**). Data are representative of three independent experiments. Mann–Whitney test; ns, non-significant; *P < 0.05; **P < 0.01

one differentiates into effector cells. However, the surrogate markers of these two subtypes are still debated. They can either be distinguished by their pattern of immunoglobulin (Ig) isotype expression or by their expression of CD80. IgM$^+$ and CD80$^-$ memory B cells remain plastic and rediversify in GC while isotype-switched and CD80$^+$ memory B cells become Ab-secreting effector plasma cells. We thus investigated whether this heterogeneity could also be found between the local and circulating memory B cells. We observed that around 80% of the memory B cells in the dLN were mainly IgG2a/b$^+$, reflecting the type 1 polarization due to IFA + CpG immunization (Fig. 5b). In contrast, most of the memory B cells in the spleen were IgM$^+$ (Fig. 5b). Moreover, the great majority of memory B cells in the dLN was CD80$^+$ while among spleen memory B cells around 40% were CD80$^-$ (Fig. 5c). Using gMFI of Ag binding at optimal labeling concentrations, we found that NP-specific memory B cells in the dLN had a higher binding than their spleen counterparts (Fig. 5d). This difference could not be attributed to a difference in BCR density since circulating memory B cells expressed more BCR than their local counterparts as estimated by Ig lambda 1 staining at the surface of the NP-specific B cells (Fig. 5d). Thus, BCR borne by local memory B cells have greater

capacities to bind their cognate ligand than the circulating ones. Overall, in parallel with memory Tfh cells, a subdivision of the memory B-cell compartment can be drawn based on localization and Ag receptor. Moreover, based on their phenotype, local memory B cells form a homogenous population programmed to likely become plasma cells after Ag boost while circulating memory B cells are more heterogeneous with some cells likely to rediversify in GC.

**Local memory Tfh cells promote early class-switch recombination**. To assess the functional consequences of the subdivision of the memory Tfh cells, we studied the cytokine production of these cells after activation. We immunized C57BL/6 mice with OVA in which naive OT-II cells were transferred the day before. Thirty days after sc immunization, dLN and spleen CD44$^+$CXCR5$^+$ OT-II cells were purified and re-stimulated in vitro. As shown in Fig 6a, no differences were found between the two memory Tfh cell subsets for IL-2 RNA expression level. In contrast, spleen memory OT-II cells after activation expressed higher level of IL-21 and IFN-γ mRNA than their dLN counterparts (Fig. 6a). Further, after a short in vitro stimulation,

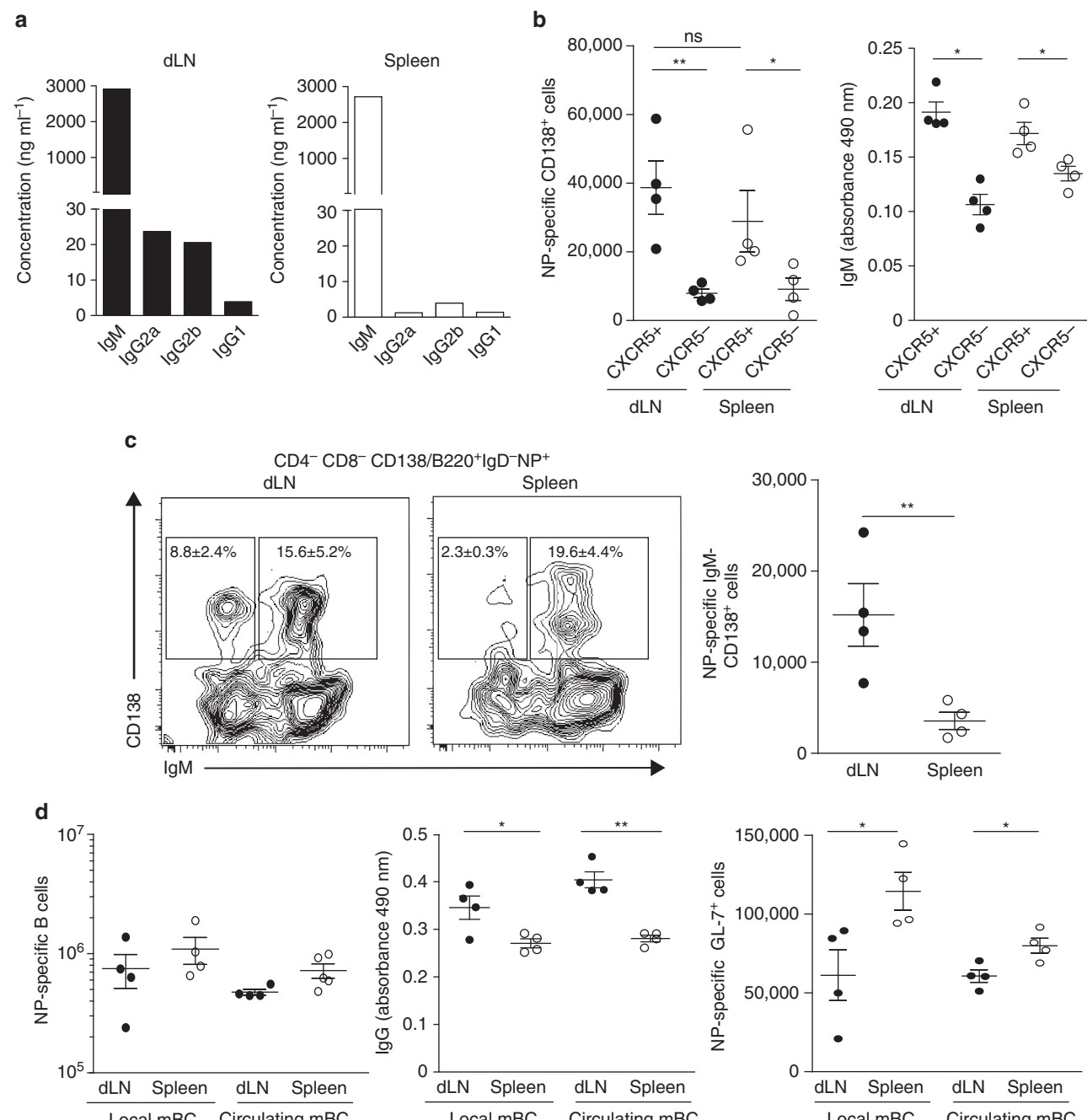

**Fig. 7** Local memory Tfh cells are more potent to induce Ig isotype-switch. Naive B cells were stimulated with anti-IgM F(ab)′₂ for 24 h and loaded with OVA peptide. Then, cells were co-cultured at ratio 1:1 with sorted dLN or spleen memory CD4⁺CD45.1⁺CD44⁺CXCR5⁺OT-II cells prepared as described in **a**. Five days later, culture supernatants were collected to perform flowcytomix experiment. Concentrations of the different Ig are presented (**a**). A number of $10^4$ dLN or spleen CD4⁺CD45.1⁺CD44⁺CXCR5⁺OT-II cells from day 30 sc immunized mice were transferred into naive mice (**b**, **c**) or into mice previously immunized 30 days before with NP-KLH in SAS either ip or sc (**d**; see Supplementary Fig. 10 for experimental scheme). Then, mice were ip immunized with NP-OVA. After 5 days, sera of mice were collected to perform ELISA and spleens were analyzed by flow cytometry. Each *dot* represents an individual mouse; *horizontal lines* denote the mean value of groups and SEM (**b**–**d**). Data are representative of three independent experiments. Mann–Whitney test; ns, non-significant; *$P < 0.05$; **$P < 0.01$

we could detect the cells that were producing IL-2, IL-21, and IFN-γ by intracellular staining (Fig. 6b). Irrespective of the three tested cytokines, we found that spleen memory OT-II cells had a higher gMFI intensity than their dLN counterparts (Fig. 6b). Thus, upon restimulation, circulating memory Tfh cells have greater capacities to produce cytokines than the local ones.

Since cytokine production is key for sustaining B-cell response, we next estimated the B-cell helper capacities of the memory Tfh cell subsets. First, naive B cells were stimulated with anti-IgM and loaded with OVA peptide. Stimulated B cells were co-cultured with purified dLN or spleen memory CD44⁺CXCR5⁺ OT-II cells. We found that both subsets of memory OT-II cells induced similar secretion of IgM by the B cells (Fig. 7a). However, Ig isotype-switch was only detected when dLN CXCR5⁺ OT-II cells were present (Fig. 7a). Thus, both local and circulating memory Tfh cells sustain IgM production in vitro. However, only the local memory Tfh cells allow early Ig class-switch recombination.

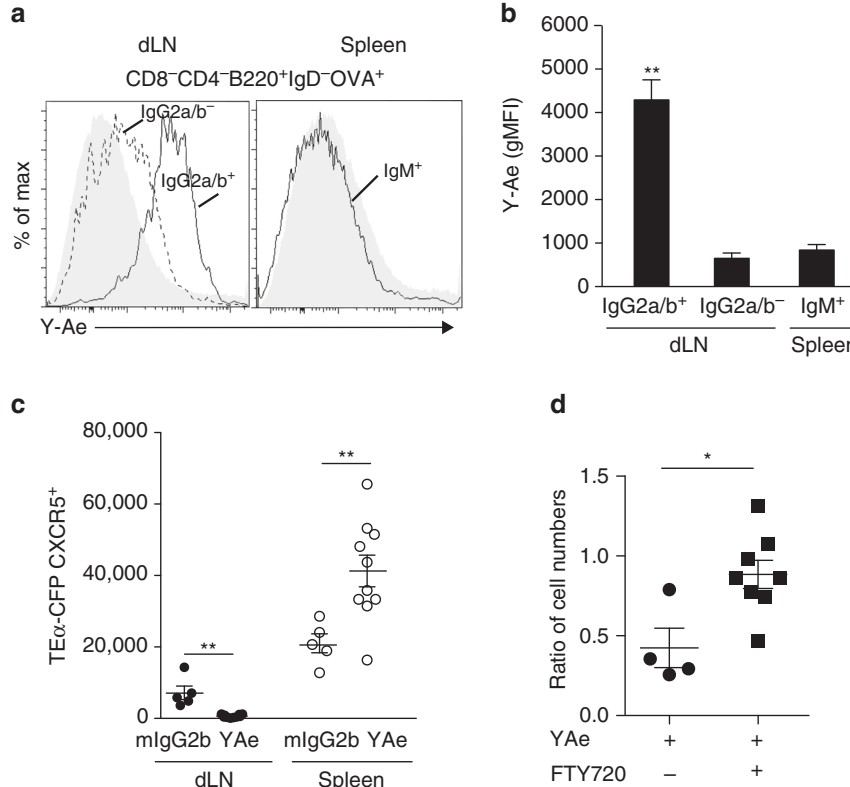

**Fig. 8** Local and circulating Ag-specific memory Tfh cells are interconnected. Histograms representing Y-Ae expression at the surface of IgG2a/b⁺ or IgG2a/b⁻IgD⁻OVA⁺ B cells in the dLN and at the surface of IgM⁺ IgD⁻OVA⁺ B cells in the spleen 30 days post immunization with 1W1K-Eα52-68-OVA in IFA + CpG. As control, mice were immunized with OVA alone (*gray* histograms) (**a**). gMFI of Y-Ae staining presented in **a** (**b**, mean ± SEM, n = 5). A number of 10⁵ CD4⁺ naive TEα-CFP were injected iv into C57BL/6 mice that were immunized with 1W1K-Eα52-68-OVA in IFA/CpG the day after. Thirty days post immunization, mice were treated every 2 days for 14 days with Y-Ae or a mIgG2b isotype control (Supplementary Fig. 11A). Number of CD4⁺CD44⁺CFP⁺CXCR5⁺ TEα cells in dLN and spleen are depicted (**c**). A number of 10⁵ CD4⁺CD45.1⁺ naive 1H3.1 cells were injected iv into C57BL/6 mice that were immunized with 1W1K-Eα52-68-OVA in IFA/CpG the day after. Thirty days post immunization, mice were treated each day for 1 week with Y-Ae or a mIgG2b isotype control and with FTY720 (10 mg kg⁻¹) or PBS as control (Supplementary Fig. 11B). Numbers of CD4⁺CD44⁺CD45.1⁺CXCR5⁺ 1H3.1 cells in dLN were enumerated and the ratios to the numbers of cells found in mice treated with isotype control and PBS are depicted for the two other experimental conditions (Y-Ae/PBS; Y-Ae/FTY720) (**d**). Each *dot* represents an individual mouse; *horizontal lines* denote the mean value of groups and SEM. Data are representative of at least three independent experiments. Mann–Whitney test; ns, non-significant; *P < 0.05; **P < 0.01

We further analyzed the in vivo helper capacities of the two subsets of memory Tfh cells. As described above, we purified dLN and spleen memory CD44⁺ OT-II cells either CXCR5⁻ or CXCR5⁺ from day 30 sc immunized mice. Each purified population was then transferred into naive C57BL/6 mice (Supplementary Fig. 10). The day after, recipient mice were ip immunized with NP-OVA and the NP-specific B-cell response was monitored 5 days after. As expected, we found that both dLN and spleen memory CXCR5⁺ OT-II cells promoted more NP-specific PC and more IgM secretion than their CXCR5⁻ counterparts (Fig. 7b). Thus, memory CXCR5+ Th cells are indeed memory Tfh cells and CXCR5 is a reliable marker to discriminate Tfh and non-Tfh cells even in the memory phase and even for the circulating memory Th cells. Strikingly, the IgM secretion was similar in mice transferred with dLN or spleen memory CXCR5⁺ OT-II cells (Fig. 7b). However, the cell count numbers of IgD⁻IgM⁻ isotype-switched PC (Fig. 7c) were far more important in mice receiving CD44⁺CXCR5⁺ OT-II cells from dLN than from spleen. Thus, local and circulating memory Tfh cells sustain the differentiation of naive B cells into plasma cells to the same level, but local memory Tfh cells promote more Ig class-switch recombination than their circulating counterparts in vitro and in vivo.

Next, to address the functional outcome of memory Tfh/memory B-cell interaction in vivo after Ag re-challenge, we immunized mice with NP-KLH to generate spleen NP-specific memory B cells either

local and circulating after ip injection and sc-immunization, respectively. These animals were then used as recipient mice in which, 30 days after immunization, local or circulating OT-II memory Tfh cells were transferred. The day after, all recipient mice received NP-OVA ip and the NP-specific B-cell response was monitored 5 days after (Supplementary Fig. 10). We found that the cell count numbers of NP-specific B cells were similar in the four experimental conditions (Fig. 7d). Thus, local and circulating memory Tfh cells sustain the early B-cell response of memory B cells to the same extent. However, the quantity of high-affinity NP-specific IgG was greater when local memory Tfh cells were transferred than when circulating ones were (Fig. 7d). In contrast, transfer of circulating memory Tfh cells induced more NP-specific B cells that were GL-7⁺ irrespective of interacting with local or circulating memory B cells (Fig. 7d). To note, we could not detect any difference in the quantity of high-affinity NP-specific IgG when local and circulating memory B cells were in front of local memory Tfh cells, nor difference in GL-7⁺ cells when local and circulating memory B cells were in front of circulating memory Tfh cells (Fig. 7d). Overall, this series of experiments demonstrates that, in vivo, local and circulating memory Tfh cells sustain the early B-cell response of naive and memory B cells but with different end products. Notably, local memory Tfh cells induce higher production of class-switched and high-affinity Ig by memory B cells while circulating memory Tfh cells induce more secondary GC.

**Circulating memory Tfh cells emerge from a local pool**. To better understand what are the molecular cues that lead to the memory Tfh cell subdivision, we tested whether local memory Tfh cells and their retention in the dLN could result from a cognate interaction with APC. To address this issue, we took advantage of Y-Ae, a mAb that specifically recognizes the pMHCII complex I-A$^b$-E$\alpha$52-68 in C57BL/6[41]. First, we found that the only Y-Ae$^+$ cells in the memory phase in the dLN were Ag-specific memory B cells (Fig. 8a). Strikingly, this was observed only for local dLN memory B cells that expressed surface IgG2a/b, which are the majority of the NP-specific B cells after IFA + CpG immunization, but not for dLN IgG2a/b$^-$ memory B cells or for spleen memory B cells (Fig. 8b). Next, we treated E$\alpha$52-68-primed mice with Y-Ae and tested whether it impacted the localization of the different memory Tfh cells (Supplementary Fig. 11A). Y-Ae treatment led to a decrease of memory Tfh cells in the dLN that correlated with an increase of spleen memory Tfh cells (Fig. 8c). Next, we tested the impact of blocking T/APC cognate interactions and lymphocyte egress at the same time. To do so, mice were treated simultaneously with Y-Ae and FTY720, a Sphingosine 1-phosphate receptor (S1PR) agonist that inhibits migration of lymphocytes[42] (Supplementary Fig. 11B). As observed before, Y-Ae treatment alone led to a decrease of memory Tfh cells remaining in the dLN showing that retention of local memory Tfh cells relies on cognate interactions (Fig. 8d). Strikingly, when we treated mice with Y-Ae and FTY720 simultaneously, we found that there were no modifications of the memory Tfh cell pool in the dLN (Fig. 8d). Thus, local memory Tfh cells are retained in the dLN through cognate interactions with pMHCII$^+$ memory B cells in the B follicle. Over time, some of the local memory Tfh cells are eventually released participating to the compartment of circulating memory Tfh cells such as the ones detected in the spleen. Overall, this demonstrates that Ag, even in the memory phase, participates to maintenance of an adequate memory anatomy, phenotype, and function.

## Discussion

In this study, we have collected evidence that the memory Tfh cell compartment can be subdivided in two distinct populations. Some memory Tfh cells are found retained in the draining lymphoid organs in the B follicle at proximity of memory B cells. In contrast, other memory Tfh cells circulate in non-draining lymphoid organs such as the spleen after sc immunization and localize outside the B follicle. Furthermore, local memory Tfh cells have a more pronounced Tfh phenotype than their circulating counterparts while both memory cells are quiescent. A similar subdivision also exists in the memory B-cell compartment. Local memory B cells express high-affinity isotype-switched Ig while circulating memory B cells are mainly low-affinity IgM$^+$. Interestingly, we found that both local and circulating memory Tfh cell subsets sustain early B-cell response after reactivation in vivo but exhibit different functions. Strikingly, long-term expression of pMHCII is detected only at the surface of Ag-specific memory B cells in the draining lymphoid organs. If cognate TCR/pMHCII interactions are abrogated, the local memory Tfh cell pool shrinks while the circulating compartment expands. Overall, this study demonstrates how memory Tfh and memory B cells are intimately intertwined. It also reveals new depths in the systemic organization and maintenance of the memory B- and T-cell compartment.

It is not unusual to divide immune memory based on the migration properties. Indeed, it has already been proposed that memory T cells can be categorized into central (TCM) and effector (TEM) memory subsets[43, 44]. CD62L$^{hi}$CCR7$^+$ TCM cells

retain the capacity to re-circulate throughout lymphoid tissue while CD62L$^{lo}$CCR7$^-$ TEM cells migrate to non-lymphoid sites and are programmed to induce inflammatory responses in situ upon Ag re-exposure. Somehow surprising, we have shown that local memory Tfh cells express mainly the CD62L$^{lo}$CCR7$^-$ TEM phenotype while they remain in lymphoid tissues, thus localization associated to TCM. Despite this phenotypic difference, we propose that local memory Tfh cells could be categorized as TEM because they remain preferentially in situ in CXCL13-rich areas where they deliver their function. In contrast, circulating memory Tfh cells express higher levels of CCR7 and re-circulate in lymphoid organs so they could be categorized as TCM. It was originally proposed that the TCM/TEM division was established by differential strength of TCR–pMHCII interactions upon recruitment of T cells into the immune response[45]. In the context of Tfh cell biology, we previously described that naive Th cells that express high-affinity TCR become preferentially effector Tfh cells[38]. Here, despite very similar public TCR repertoire, we have demonstrated that local memory Tfh express TCR with higher affinity than the circulating ones. As such, our findings parallel previous studies, including ours, that also showed these differences in TCR affinity when comparing resident vs. circulating effector or memory Th cells[1, 29]. Notably, we also demonstrated this subdivision in localization and binding capacities of the Ag-receptor to its cognate ligand for the memory B-cell compartment.

The memory Tfh cell subdivision relies on memory Tfh cell localization, but whether this has biological consequences remained unknown. We have been able to show that the phenotype of memory Tfh cell subsets is different. Local memory Tfh cells exhibit a more polarized phenotype than their circulating counterparts with higher expression levels of Bcl-6, CXCR5, ICOS, and PD-1. Moreover, around two third of the local memory Tfh cells are CD69$^+$, even day 120 after immunization, while almost no circulating memory Tfh cells express CD69. Strikingly, when we transferred local memory Tfh cells in absence of Ag in naive hosts, we observed a decrease of CXCR5 expression at the surface of the transferred cells with time. Because Bcl-6 expression is not stable and requires continuous reinforcement[30], this difference in Bcl-6 expression levels could be explained by the fact that local memory Tfh cells bear high-affinity TCR and are in close contact with pMHCII$^+$ memory B cells. Still, the Bcl-6 expression levels are lower to the ones found for effector Tfh cells, which is in accordance with previous studies showing that a cell, after leaving a GC, acquires a less Tfh-polarized phenotype and Bcl-6 expression is reduced[23, 25, 46]. Unlike MHCI-restricted CD8 T cells, it is now clear that Th cells require persistent Ag to achieve maximal expansion in vivo[47]. There is also evidence for pMHCII expression with detectable impact on Th cells after viral infection[48] or after protein immunization[29]. Furthermore, follicular dendritic cells acquire immune complexes and can retain and display them periodically on their cell surface for extended periods[49]. Recently, it was even shown using two-photon microscopy that memory Tfh cells scan subscapular sinus macrophages for Ag[50]. Hence, we reasoned that pMHCII expression could trap local memory Tfh cells that would eventually result in their more Tfh-polarized phenotype as compared to circulating ones. We indeed detected pMHCII at the surface of cells in the dLN in the memory phase, but only Ag-specific memory B cells. Moreover, in vivo blocking of pMHCII/TCR interactions correlated with the release of local memory Tfh cells into the circulating compartment. Taken together our studies therefore suggest that pMHCII depots could act as a local mechanism that shapes the TCR repertoire of memory Tfh cells such as cells bearing high-affinity TCR are retained and those with lower affinities are released. The competition for pMHCII/TCR cognate interaction

would participate to this phenomenon since only few pMHCII[+] B cells can be found in the memory phase. While Ag is not required for the maintenance of immune memory[51–53], we propose that Ag is important for the correct placement of memory Tfh cells in proximity to memory B cells in the B follicles of the lymphoid site draining initial Ag entry with the two memory cell compartments perceiving this interaction as memory sustaining, as opposed to an immune challenge resulting in activation and effector outcomes. This would not be peculiar to memory Tfh cells since it parallels the formation of a peripheral cellular niche of tissue-resident memory CD8[+] T cells (TRM) in non-lymphoid organs. These cells, as for local memory Tfh cells, express CD69, which seems to act as an additional retention signal on TRM[54, 55]. Moreover, it has been shown very recently that transient expression of Ag impacts TRM generation and that Ag-dependent competition shapes the repertoire of TRM[56, 57]. Furthermore, in favor of this hypothesis, a recent study showed that Rituximab treatment resulted in a lack of naive and GC B cells in human lymph nodes without affecting the Tfh cell populations[58]. Thus, Tfh cells do not require an ongoing GC response for their maintenance. However, what the authors also showed is that Rituximab treatment had no effect on the pool of memory B cells thus confirming that memory Tfh/memory B cell interactions control the maintenance of the pool of memory lymphocytes.

While our work focused on the phenotype and function of memory Tfh cells, we also studied the phenotype of memory B cells. We found that memory B-cell phenotype can also be categorized based on their localization. Local memory B cells form a homogeneous population, bear class-switched Ig and express a phenotype that likely programs them to rapidly become Ab-secreting plasma cells after Ag re-exposure. In contrast, circulating memory B cells bear mainly IgM and form a heterogeneous population phenotypically, with around one third of these cells that potentially has the capacity to rediversify in GC. We further tested the functional outcome of these phenotypic differences. Using cell transfer, we were able to address in vivo how local and circulating memory Tfh cells regulate the early B-cell response of naive and memory B cells. We comprehensively demonstrated that both subsets of memory Tfh cells have the capacity to sustain an emerging or a secondary B-cell response to a level that is far higher than their CXCR5[−] counterparts. Strikingly, local memory Tfh cells promote more plasma cell differentiation. In contrast, circulating memory Tfh cells secrete more IFN-γ and IL-21 and are more prone to support GC than local ones. These findings were unexpected. Indeed, it has been previously shown that the B-cell help provided by CXCR3[+]CXCR5[+] circulating Th cells from human blood was very weak in vitro after super-Ag stimulation[20, 21]. However, here we show that circulating memory Tfh cells are mainly PD-1[lo]CXCR3[+] and are localized outside of the B follicle. One explanation could be that IFA complemented with CpG favors the development of type 1-immunity and of IgG2a class switch induced by IFN-γ, thus the Ag-specific circulating memory Tfh cells that we track express CXCR3 and are the cells with B-cell helper capacities in vivo. We therefore propose that given the fact that the circulating B-cell memory is more plastic, this organization would ultimately enhance durable immune protection. One other interesting finding was that this subdivision is not peculiar to protein vaccination but was also observed after virus infection such as Influenza PR8 intranasal infection. Altogether, our studies therefore emphasize how the systemic memory T- and B-cell compartment is organized to efficiently protect against a similar or related Ag that primed in the first place this memory pool.

One other question raised by our studies is how this memory anatomy develops. Published observations favor an early

programming during the immune response to obtain this organization. Asymmetric segregation during T lymphocyte proliferation after activation would lead to different cell fates[59]. Moreover, a comprehensive analysis of Th cell response over the primary and memory responses in LN demonstrated the requirement for co-stimulation at the original priming for the development of effective memory responses[60]. In addition, effector Tfh cells develop only locally in the draining lymphoid organs[1]. Finally, He et al.[24] showed that circulating CCR7[lo]PD-1[hi]CXCR5[+] Th cells are generated early during Tfh differentiation and participate to the emergence of the memory pool. Here we show that the development of the memory compartment is not only programmed early but is a long-term process, and demonstrated that local and circulating memory Tfh cells are interconnected. Importantly, this process could be critical for effective protein vaccination and required to promote a rapid and robust secondary response to the vaccine boost. In this manner, as local pMHCII expression would decline with time, the local and circulating Tfh compartments would also decay.

In summary, our results clearly indicate that memory Tfh and memory B cells are intimately connected. In the draining lymphoid organs, both memory Tfh and memory B cells interact in the B follicle to form a pool that serves as a reservoir. Over time, some cells are released from this pool and participate to the circulating memory compartment. This circulating population is highly effective and plastic allowing the possibility to these cells to rapidly react and adapt to Ag re-exposure. Overall, this specific memory anatomy would ultimately allow a better immune protection. Hence, creating and boosting local depots of Ag in a form of pMHCII and/or enhancing memory Tfh cell compartment by addition of CpG, the TLR9 agonist, to the vaccine emulsion[37] become important mechanisms for controlling local immunity and, more importantly, regulating the systemic and highly protective memory compartment.

## Methods

**Mice**. C57BL/6 (CD45.2[+]) mice were purchased from Centre d'Elevage Janvier (Genest Saint Isle, France). C57BL/6 OT-II (CD45.2[+]CD45.1[+]) TCR transgenic mice, C57BL/6 TEα TCR transgenic mice expressing CFP under the actin promoter (TEa-CFP), and C57BL/6 1H3.1 (CD45.2[+]CD45.1[+]) TCR transgenic mice were bred in UMS006. Only females of 8–12 weeks of age were used for experimental procedures. All experiments were performed in accordance with the national and European regulations and institutional guidelines. Mouse experimental protocols were approved by the French 'Ministère de l'Enseignement Supérieur et de la Recherche' (ethical review no. MP/19/58/06/12).

**Immunization and reagents**. IFA, Sigma Adjuvant System (SAS) and OVA protein were from Sigma-Aldrich. CpG (5′-TCCATGACGTTCCTGACGTT-3′) was from Miltenyi Biotec. OVA peptide (ISQAVHAAHAEINEAGR), 1W1K (EAWGALANKAVDKA)-OVA, and 1W1K-Eα52-68 (ASFEAQGALANIAVD KA)-OVA from Genecust. NP-OVA and NP-KLH from Biosearch Technologies. Mice were either immunized sc at the base of tail or ip with 100 µg of 1W1K-OVA, 1W1K- Eα52-68-OVA, OVA, NP-OVA or NP-KLH in the indicated adjuvant. FTY720 was used at the concentration of 10 mg kg[−1] (Cayman Chemical).

**In vitro culture**. Cell cultures were set up in a medium consisting of RPMI 1640 supplemented with 10% FCS, 10 mM L-glutamine, 100 U ml[−1] penicillin, and 100 µg ml[−1] streptomycin (from Life Technologies BRL Life Technologies) in 24- or 96-well flat-bottom culture plates in a final volume of 1 ml and 250 µl, respectively. Supernatants were collected to perform Mouse Immunoglobulin Isotyping flowcytomix experiment following the manufacturers' protocol (eBioscience).

**ELISA**. ELISA plates (Thermo Scientific) were coated with 10 µg ml[−1] NP15-BSA (Biosearch Technologies Inc.). NP-specific IgM and IgG were detected in plasma from blood by ELISA[37].

**RNA extraction, cDNA synthesis, and quantitative PCR**. RNA was isolated with the RNeasy Mini Kit (Quiagen). cDNA was reverse transcribed with the High-Capacity cDNA Reverse Transcription Kit (Applied Biosystems).

Quantitative PCR amplifications were performed using TaqMan Universal PCRMaster Mix (Applied Biosystems) or SYBR GREEN I Master (Roche Applied Science) and were performed on the LightCycler 480 (Roche Applied Science; see Supplementary Table 1 for primers).

**Immunoscope analyses.** PCR was conducted in 50 μl on 1/50 of the cDNA with 2 U of Taq polymerase (Promega) in the supplier's buffer. cDNA was amplified using Vβ-specific sense primers and antisense primers hybridizing in Cβ segments (Supplementary Table 1). Amplified products were then used as template for an elongation reaction with fluorescent-tagged oligonucleotides (run-off reactions).

**Cloning and sequencing of TCRBV rearrangements.** TOPO Blunt cloning kit (Invitrogen Life Technologies) was used. Then, PCR amplification was performed and was followed by a second step of elongation using an ABI PRISM Big DyeTerminator kit (Applied Biosystems). Sequencing products were then read on 16 capillaries (Genetic Analyzer; Applied Biosystems).

**Flow cytometry analysis and cell sorting.** Cell suspensions were prepared in PBS/2% FCS, 5 mM EDTA. For Ag-specific Th and B cells analysis, organs were dissociated, filtered and treated with 2.4G2 for 10 min. To track antigen-specific CD4+ T cells, cells were incubated with PE-1W1K-IA$^b$ tetramer (7 μg ml$^{-1}$) and APC anti-CXCR5 (REA 215, Miltenyi Biotec, 1:50) for 2 h at room temperature. The tetramer 1W1K-IA$^b$ is obtained from NIH Tetramer core facility. To track, antigen-specific B cells were stained for 60 min with NP-PE (Biosearch Technologies Inc.) or OVA-Alexa488 (Invitrogen) at a final concentration of 1 μg ml$^{-1}$. After tetramer or NP-PE staining, cells were washed and then incubated on ice for 45 min with fluorophore (or biotin)-labeled mAbs. The following mAbs purchased from BD Biosciences were used: anti-Bcl-6 (K112-91, 1:50), anti-CCR7 (4B12, 1:200), anti-CD69 (H1.2F3, 1:100), anti-CXCR3 (CXCR3-173, 1:100), anti-IgM (II/41, 1:200), anti-IgG2a/b (R2-40, 1:100), anti-Vβ14 (14-2, 1:100), anti-TCRβ (H57 −597, 1:100), anti-Ki67 (SolA15, 1:100), anti-BrdU (3D4, 1:100), anti-CD138 (281-2, 1:200), anti-CD45.1 (A20, 1:100), anti-CD4 (RM4-5, 1:200), anti-CD8α (53-6.7, 1:500), and anti-CD95 (15A7, 1:100). The following mAbs purchased from eBioscience were used: anti-B220 (RA3-6B2, 1:200), anti-CD4 (RM4-5, 1:200), anti-Ly-6C (AL-21i, 1:200), anti-GL-7 (GL-7, 1/100), anti-CD62L (MEL-14, 1:200), anti-CD44 (IM7, 1:400), anti-CCR6 (29-2L17, 1:50), anti-IgD (11-26c, 1:500), anti-ICOS (7E.17G9, 1:200), anti-PD-1 (J43, 1:200), anti-CD80 (16-10OA1, 1:100), anti-PDL-2 (TY25, 1:100), anti-IL2 (JES6-5H4, 1:100), anti-IL21 (FFA21, 1:50), anti-IFNγ ()XMG1.2, 1:100), and biotin-Y-Ae (ebioY-Ae, 1:400). The cells were then suspended with Fixable Viability Dye eFluor450 or eFluor660 (eBioscience) for dead cells exclusion. For intracellular staining, cell suspensions were fixed and permabilized using BD Fixation/Permeabilization kit. For intracellular cytokine staining, sorted cells were stimulated with 50 ng ml$^{-1}$ PMA (Sigma) and 1.5 μg ml$^{-1}$ ionomycin (Sigma) in the presence of 10 μg ml$^{-1}$ brefeldin A (eBioscience) and 2 μM monensin (eBioscience) for 4 h, followed by surface and intracellular staining. Data were collected on a BD LSRII™ or on a BD LSRII/Fortessa (BD Biosciences) and analyzed using FlowJo software (Tree Star). To generate antigen-specific memory T cells, naive transgenic T cells were transferred intravenously (iv) into C57BL/6 mice, followed by sc immunization. At day 30 post immunization, mice were killed; dLN (inguinal and periaortic) and spleen were harvested for sorting of local and circulating memory Tfh cells respectively using FACSARIA-SORP (BD Biosciences).

**Immunofluorescence.** dLN and spleen were harvested into PLP buffer (0.05 M phosphate buffer containing 0.2 mL lysine (pH 7.4), 2 mg ml$^{-1}$ NaIO$_4$, 10 mg ml$^{-1}$ paraformaldehyde), fixed overnight and dehydrated in 30% sucrose prior to embedding in OCT freezing media (Sakura Finetteck). Frozen sections (10 μm) were cut on a CM1950 Cryostat. Sections were stained in PBS (0.01% Triton X-100 and 5% goat serum) using the following Abs: anti-B220 (RA3-6B2, 1:200, eBioscience) and anti-CD4 (RM4-5, 1:200, eBioscience), CD45.1 (A20, 1:100, BD Biosciences). Images were acquired on a Apotome ZEISS Inv. Regions and cells were defined with IMARIS image analysis software.

**Statistical analysis.** Differences between variables were evaluated using the non-parametric Mann–Whitney test. All statistical analyses were carried out with Prism 4.0 software (GraphPad). P-values <0.05 were considered statistically different.

**Data availability.** The data that support the findings of this study are available from the corresponding author upon request.

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

## Acknowledgements

We thank M. Ducatez for providing NP8 strain, M. Linterman, A. Dejean, S. Laffont-Pradines, A. Sacquin, and S. Guerder for reviewing the manuscript and all the members of the Guerder team for helpful comments on the manuscript. We also thank the flow cytometry facility (F. L'Faqihi, V. Duplan, and A.L. Iscache), the animal house staff members (INSERM UMS06), the microscopy core (A. Canivet and S. Allart), the sequencing core (UDEAR CNRS Toulouse), and the NIH Tetramer Facility. This work was supported by Le Conseil Régional Midi-Pyrénées, Institut National contre le Cancer (INCa, PLBIO10-195, and INCA-6530) and ANR (EQUIPEX, ANR-16-CE15-0019-02, and ANR-16-CE15-0002-02).

## Author contributions

A.A. and N.F. conceived and designed the studies. A.A., M.A., M.G. and C.P. carried out experiments. A.A. and N.F. wrote the manuscript with contributions from co-authors.

## Additional information

**Competing interests:** The authors declare no competing financial interests.

