## [Peer review file · Nature Communications]

Reviewers' comments:

Reviewer #1 (Remarks to the Author):

Similar to other T cell subsets, Tfh cells can differentiate into memory cells. Previous studies show that memory Tfh cells persist in the draining LNs, or become a circulating compartment which resembles to central memory cells. Asrir et al analyzed the biological similarities and differences between LN-resident (called local in the paper) memory Tfh cells and circulating memory Tfh cells. The group showed that local memory Tfh cells were with TCRs of higher affinities than circulating memory Tfh cells. Local memory Tfh cells also expressed higher levels of CXCR5, Bcl-6, ICOS, and PD-1, but lower levels of CCR7. These observations were confirmed in different models including OVA-OT-II model, immunizations with different adjuvants, and influenza infection. While local memory Tfh cells remained within follicles, circulating memory Tfh cells were primarily found outside follicles. Analysis of TCR clonotypes showed that the two compartments shared the same clonotypes, suggesting that circulating memory Tfh cells originated from local Tfh cells. Last, memory B cells in draining LNs expressed the peptide-class II complex for long term, and the treatment of mice with an mAb which interruptus the interactions between the peptide-class II complex and TCR increased the number of circulating memory Tfh cells. This suggests that high affinity local memory Tfh cells were trapped within the draining LNs through interactions with memory B cells presenting antigens. Whereas the same concept was already suggested by the senior author almost a decade ago (Fazzileau et al, NI, 2007), this study provides additional important observations regarding the differences in biological features between local and circulating memory Tfh cells. However, the other major part of this study, which is on the analysis of local memory B cells and circulating memory B cells, is less convincing. First, the analysis of NP-specific memory B cells was totally dependent on the detection by flow of cells which bound to NP-PE. There was no evidence proving that NP-PE+ cells in the gating of Fig. 5a were really antigen-specific. Thus, I am not sure whether the observations in 5b-d were correct. The optimal would be to expand the gated B cells and analyze the produced Ig, but the authors at least should provide the data with unimmunized mice, or immunized with different antigens, to show the extent of background staining. Second, I do not entirely agree with the interpretation of Fig. 5c, and the difference in the biological roles between local and circulating memory B cells. In Fig. 5c, while splenic NP+ memory B cells showed a bimodal CD80 expression pattern (also given that NP+ gating was correct), dLN NP+ memory B cells showed one peak, and the global CD80 expression was lower than CD80+ splenic B cells. If the gate for negatives was applied in a same way as in splenic cells, approximately half of dLN memory B cells would be negative for CD80. How was PD-L2 expression? The authors proposed that "local memory B cells were programmed to become plasma cells while circulating memory B cells were heterogeneous with cells capable of redifferentiation", but this was not supported in the experiments in Fig. 7d. If the hypothesis is correct, I would expect that NP-specific IgG was higher in local mBC+dLN Tfh than in circulating mBC+dLN Tfh, but this was not the case. I agree that there was a tendency that local memory Tfh cells induce more class-switching and circulating memory Tfh cells induce more GC B cell differentiation, but I am not sure if this is "highly intertwined" with memory B cells. I suggest to soften the conclusions on the part of memory B cells.

Other points:

1. I do not quite understand why the global TCR affinity was much higher in local memory Tfh cells, although dominant clonotypes were shared between local and circulating Tfh cells. Is there evidence that the identified clonotypes are "public" and of high affinity?
2. It is puzzling why circulating memory Tfh cells expressed higher levels of cytokines upon activation. Was there any difference in activation-induced cell death? How was Fas expression?
3. In Fig. 7, B cell phenotype was analyzed at day 5, which is very early for the detection of class-switched B cells in naïve mice. Did the authors look at later time points?

Reviewer #2 (Remarks to the Author):

The manuscript of Asrir and colleagues attempts to address an interesting question: the relationship between the site of the immune response and the location and function of subsequent memory cells. The authors propose that memory T follicular helper (mTfh) cells can be divided into a local populations, that have the hallmarks of Tfh cells and are of marginally higher affinity for the antigen, and “circulating” population assayed in the spleen that is also antigen specific but lacks most Tfh attributes. The memory B cells are also proposed to have similar anatomical demarcation, with local memory B cells being predisposed to PC differentiation, while circulating memory can re-enter germinal centers. Finally the authors provide evidence that the local memory Tfh is held in the draining lymph nodes by cognate antigen-MHCII interaction with antigen specific memory B cells.

Overall this is an ambitious study that uses some nice experimental approaches to test the memory model outlined above. Although it is well known that Tfh cells can form memory and that a circulating memory population is described, the novelty here is that the authors attempt to provide are more holistic model that incorporates the anatomical sites of the immune response. However I feel that the approaches currently taken, although examined exhaustively in the manuscript, are, as outlined below, not sufficient to convincingly prove the model being put forward.

Specific comments.

Major points.

1. The relationship of the local and circulating memory Tfh populations is not convincingly shown. The spleen mTfh cells while expressing a somewhat lower level of CXCR5 (fig 2b), express the same amount of Bcl6 as CXCR5-negative cells in the dLN and low levels of PD1 and ICOS. What is the authors' evidence that this population does not include other populations such as Th1 type memory cells? Sorting, transfer and long term monitoring of effector Tfh, local and circulating memory Tfh cells might shed light on the relationship between the cell subset.
2. The authors assume that there is no residual GC activity in the dLN at day 30, and that GCs were not initiated in the spleen after sc immunization and flu infection. Yet their own data show that >20% of cells are ki67+ and BrdU+ at day 30 (fig1c-d). In this reviewers experience there are residual GCs in both immunization and after flu infection after 4 weeks and that some GC activity can occur (especially in flu infection) in a distal site such as the spleen, so this issue needs to be carefully examined, by immunofluorescence or more quantitatively by FACS for GC B cells.
3. Background. The authors do not show the level of binding of the 1W1K tetramer to non-immunized mice at any point. This is important to show specificity, especially in the situations such as the spleen after sc immunization where the frequencies of antigen specific cells are low (and in Suppl figures).
4. The identification of memory B cells is not sufficient. The more standard approach is to have a dump channel using anti-Gr1 to remove antibody-antigen binding myeloid cells that are especially as issue in the spleen. Moreover CD38 is used to separate antigen-specific memory from GC B cells that are not assessed. Similarly as outlined above the authors should show some non-immunized mice as background staining, this is most pertinent to determine if the NP-low binding cells in fig 5a spleen are indeed antigen specific.
5. Figure 6. The differences in cytokine production between the restimulated dLN and spleen populations seems very marginal at best as depicted from panels b and c. The fold change shown in panel a does not appear to be an accurate representation of the small differences and should be removed. The cytokine production should be quantified using and independent method such as real time PCR. Even if the small differences in cytokine production are real, the data in fig 7 shows that the lower cytokine producing dLN mTfh cells are better at supporting class switching, a finding that raises doubt on any interpretation of fig 6.
6. Following on from point #5, the mechanism by which the local and circulating mTfh cells would induce GC in the spleen and IgG PCs in the dLN is unclear and a major limitation of the paper. The authors propose a role for IL21 in promoting the GCs, yet this disregards the very potent function

of IL21 inducing PC differentiation and the lack of extrafollicular plasmablasts in the IL21 KO mice after immunization. Moreover the function of IL21 in GCs is only apparent late after immunization, not at the early stages examined here (ref 9, 10)

7. Model. The overall conclusion is that mTfh cells are held in the local dLN by cognate peptide-MHCII interactions with memory B cells. For this to be the case the antigen specific Tfh and memory B cells (presumably affinity matured) need to perceive this interaction as memory sustaining, as opposed to an immune challenge resulting in activation and effector outcomes such as PC differentiation. The authors do not provide a convincing model to explain how the cells differentially control these outcomes, given that this is a antigen-specific interaction.

Other issues.

8. The experiments and figures are complex (transfer, TCR Tg, different routes of challenge etc) making some aspects difficult to read and compare. To help the reader the authors should consider providing some schema for the experimental flow on the figures associated with some of the more complex experiments.

9. The data in Fig 3 show on the whole that the Vb gene usage is very similar between the antigen-specific local and circulating Tfh cells. However the RT-PCR in panel d (showing very similar Vb14 %) does not match the FACS estimation in panel e showing a pronounced difference between spleen and dLN. One of these measurements must be inaccurate.

10. It is mentioned a couple of times in the text, that the memory B cells and high affinity, but this is not actually shown, and only NP15 (total NP) is measured in the ELISAs.

We would like to thank the Reviewers for their contributions to this study and hope that they feel the changes added to the revised manuscript in response to their queries have substantially improved the quality of the paper, as we do. We were encouraged by the Reviewers' interest and comments regarding the original manuscript and have modified our manuscript to address their concerns and strengthen it. The scope and conclusions of the original manuscript remain largely unchanged but the manuscript is clearer and more precise. A point-by-point response to each of the Reviewers' comments is provided below.

Responses to Reviewer#1

In general, Reviewer 1 was positive about the findings presented in the original manuscript. The first Reviewer highlighted the novelty of the findings and stated that 'this study provides additional important observations regarding the differences in biological features between local and circulating memory Tfh cells'.

Comments

'However, the other major part of this study, which is on the analysis of local memory B cells and circulating memory B cells, is less convincing. First, the analysis of NP-specific memory B cells was totally dependent on the detection by flow of cells which bound to NP-PE. There was no evidence proving that NP-PE+ cells in the gating of Fig. 5a were really antigen-specific. Thus, I am not sure whether the observations in 5b-d were correct. The optimal would be to expand the gated B cells and analyze the produced Ig, but the authors at least should provide the data with unimmunized mice, or immunized with different antigens, to show the extent of background staining.'

Response: We indeed did not present any control staining in the original version of the manuscript as the use of labelled NP to detect NP-specific B cells is now well documented. However, as depicted in Figure 1 for Reviewers (Fig R1), we have established the use of NP-PE in unimmunized mice or in mice immunized with an irrelevant protein (OVA) emulsified in IFA/CpG in which almost no NP-specific B cells can be detected in draining LN and in spleen.

Action: We have modified Figure 5 of the revised manuscript using the data presented in Fig R1 to show that the NP-PE gate is accurate and have modified the text accordingly page 11 and page 35.

'Second, I do not entirely agree with the interpretation of Fig. 5c, and the difference in the biological roles between local and circulating memory B cells. In Fig. 5c, while splenic NP+ memory B cells showed a bimodal CD80 expression pattern (also given that NP+ gating was correct), dLN NP+ memory B cells showed one peak, and the global CD80 expression was lower than CD80+ splenic B cells. If the gate for negatives was applied in a same way as in splenic cells, approximately half of dLN memory B cells would be negative for CD80. How was PD-L2 expression?'

Response: As suggested by this reviewer comment, we have monitored the geometric mean fluorescence intensity (gMFI) of CD80 staining at the surface of the draining LN and spleen NP⁺ memory B cell subsets and we indeed observed a significant difference (depicted in figure 2 for reviewers, Fig R2, panel a). Anyhow, if we compare CD80 expression at the surface of these cells as compared to an isotype control, we clearly observe that the frequency of CD80 negative cells is significantly lower in draining LN NP⁺ memory B cells as compared to splenic NP⁺ memory B cells (Fig R2, panel b). Actually, we observed that even the expression level of the isotype control was lower at the surface of draining LN cells as compared to the one at the surface of spleen cells (gMFI : 35.1 vs 102). We therefore think that we see a bimodal expression of CD80 at the surface of spleen cells because we have two distinct populations (CD80^{neg} and CD80⁺) while the CD80 expression at the surface of draining LN cells was almost unimodal because there was so few CD80^{neg} cells and almost only CD80⁺ cells. Regarding PD-L2 expression, as depicted in figure 3 for Reviewers (Fig R3), we can observe that the great majority the NP⁺ memory B cells that are CD80^{neg} are PD-L2^{neg} too, irrespective of whether they are from draining LN or from spleen.

Action: This is true that the overall CD80 expression at the surface of dLN NP⁺ memory B cells is lower than splenic NP⁺ memory B cells. Anyhow, this is also true that the frequency of CD80^{neg} NP⁺ memory B cells is greater in the spleen than in the dLN. Overall, our original conclusion remains the same, thus we did not decided to modify the revised manuscript accordingly. Regarding PD-L2 expression, Fig R3 can be added to the main figure or as a supplementary figure if deemed necessary.

'The authors proposed that "local memory B cells were programmed to become plasma cells while circulating memory B cells were heterogeneous with cells capable of rediversification", but this was not supported in the experiments in Fig. 7d. If the hypothesis is correct, I would expect that NP-specific IgG was higher in local mBC+dLN Tfh than in circulating mBC+dLN Tfh, but this was not the case. I agree that there was a tendency that local memory Tfh cells induce more class-switching and circulating memory Tfh cells induce more GC B cell differentiation, but I am not sure if this is "highly intertwined" with memory B cells. I suggest to soften the conclusions on the part of memory B cells.'

Response: We agree with this reviewer comment and have modified the text page 2 (abstract), page 5 and page 14 to soften the conclusion on the part of memory B cells.

'1. I do not quite understand why the global TCR affinity was much higher in local memory Tfh cells, although dominant clonotypes were shared between local and circulating Tfh cells. Is there evidence that the identified clonotypes are "public" and of high affinity?'

Response: Public repertoire was first proposed by Cibotti *et al* in a seminal paper published in J Exp Med in 1994 (Cibotti *et al.* 1994). This paper proposed the following definition: public is identical in all individual, private is specific to each individual. Since then, this terminology has been largely used in T cell repertoire studies and the notions of public or private TCR repertoire are now common knowledge in the field. Here, we found that the TCR repertoire used by local and circulating 1W1K-specific mTfh largely overlapped. Indeed, we found that the dominant repertoire was similar between draining LN and spleen mTfh cell subsets. The CDR3 β of the dominant clonotypes were shared by all individuals and found in both cell subsets. Thus, these dominant clonotypes can be classified as 'public'. The repertoire difference found between the two cell subsets can therefore not be accounted to the public and dominant repertoire but to the private repertoire. These private clonotypes, by definition, are distinct between individuals, thus remain almost impossible to study. Regarding the affinity of the 'public' clonotypes, we have currently in hands no measurement of it and therefore cannot directly answer to this reviewer question.

Action: Based on this reviewer comment, we think that it was not clear in the original version of the manuscript and have clarified this point in the revised version of the manuscript in page 10.

'2. It is puzzling why circulating memory Tfh cells expressed higher levels of cytokines upon activation. Was there any difference in activation-induced cell death? How was Fas expression?'

Response: Based on this reviewer comment, we analyzed back the flow cytometry experiments corresponding to Figure 6b-c. In the first place, we unfortunately did not assess Fas expression nor activation-induced cell death. Anyhow, we were able to monitor in these experiments the frequency of dead cells since we used a viability dye. As shown in figure 4 for Reviewers (Fig R4), we can see that the frequency of activated transgenic cells (CD45.1⁺ cells) that were dead (viability dye positive) was similar between dLN and spleen cells.

Action: Fig R4 can be added as a supplementary figure if deemed necessary.

'3. In Fig. 7, B cell phenotype was analyzed at day 5, which is very early for the detection of class-switched B cells in naive mice. Did the authors look at later time points?'

Response: We indeed made the choice to look to the NP⁺ B cell response very early after immunization. This choice was made based on the literature that clearly shows that the peak of the secondary NP-specific B cell response is reached as of day 4 post-boost (for example see McHeyzer-Williams *et al.*, J Exp Med, 2000). Since our aim was to be the closest to the physiology and since we studied the impact of memory cells, we therefore decided to look 5 days post-immunization.

Responses to Reviewer#2

This reviewer was also positive about the novelty of the original manuscript since Reviewer 2 stated 'the novelty is that the authors attempt to provide a more holistic model that incorporates the anatomical sites of the immune response'. Moreover, Reviewer 2 also emphasized that 'this is an ambitious study that uses some nice experimental approaches to test the memory model.'

1. The relationship of the local and circulating memory Tfh populations is not convincingly shown. The spleen mTfh cells while expressing a somewhat lower level of CXCR5 (fig 2b), express the same amount of Bcl6 as CXCR5-negative cells in the dLN and low levels of PD1 and ICOS. What is the authors' evidence that this population does not include other populations such as Th1 type memory cells? Sorting, transfer and long term monitoring of effector Tfh, local and circulating memory Tfh cells might shed light on the relationship between the cell subset.

Response: Discriminating 1W1K-specific memory Th cells based on CXCR5 expression could have been problematic. Anyhow, as depicted in Figure 7b of the original manuscript, we were able to show that dLN and spleen memory Th cells that were CXCR5⁺ promoted more NP-specific PC and more IgM secretion than their CXCR5^{neg} counterparts. Thus, even if spleen memory Th cells express Bcl6 to the same level than CXCR5^{neg} dLN memory Th cells, their functional capacity as a B cell helper was greater than CXCR5^{neg} dLN memory Th cells. Therefore, as stated in the text, we think that 'memory CXCR5⁺ Th cells are indeed mTfh and CXCR5 is a reliable marker to discriminate Tfh and non-Tfh cells even in the memory phase and even for the circulating memory Th cells'.

2. The authors assume that there is no residual GC activity in the dLN at day 30, and that GCs were not initiated in the spleen after sc immunization and flu infection. Yet their own data show that >20% of cells are ki67+ and BrdU+ at day 30 (fgi1c-d). In this reviewers experience there are residual GCs in both immunization and after flu infection after 4 weeks and that some GC activity can occur (especially in flu infection) in a distal site such as the spleen, so this issue needs to be carefully examined, by immunofluorescence or more quantitatively by FACS for GC B cells.

Response: This is a fair comment that we have assessed in different ways. First of all, this is true that we did not present in the first place the phenotype of the NP⁺ B cells after immunization. As depicted in figure 5 for reviewers (Fig R5), we assessed the phenotype of these cells by looking at GL-7 and CD38 expression and we found GL-7⁺CD38⁻ GC B cells in the draining LN but not in the spleen at day 14 post-immunization, thus the GC reaction occurred only locally in the dLN after sc immunization. Strikingly, the GC reaction was ended by day 30 post-immunization since only around 20% of NP⁺ cells were GL-7⁺CD38⁻ at this later time point at the frequency and cell count number levels (Fig R5). Second, as suggested by this reviewer, we also performed immunofluorescence and observed that GC activity, monitored as GL-7⁺ structures in B follicles (IgD⁺) per mm², was maximum in dLN at day 14 post-immunization as compared to unimmunized or day 30 post-immunized mice. More importantly, GC activity was non significantly different in unimmunized mice and in day 30 post-immunized mice. These results are depicted in figure 6 for Reviewers (Fig R6). Finally, in figure 7 for Reviewers (Fig R7), we have depicted the intracellular level of Ki-67 in CXCR5⁺ Th cells from the draining LN or the spleen of immunized mice. We observed that this percentage is indeed around 20% but that this frequency was stable with time post-immunization when comparing day 30 to day 90 sc immunized mice. To note, regarding the

data on flu infection presented in Fig S5 of the original manuscript (now Fig S6), all the experiments were performed at day 45 but not at day 30 post-infection.

Action: We have added two supplementary figures (Fig S8 and S9) using the data presented in Fig R5 and Fig R6 to clearly show that we track NP⁺ memory B cells both in the draining LN and in the spleen at day 30 post-immunization and have modified the text accordingly page 11. Regarding Fig R7, this figure can be added as supplementary figure if deemed necessary.

3. Background. The authors do not show the level of binding of the 1W1K tetramer to non-immunized mice at any point. This is important to show specificity, especially in the situations such as the spleen after sc immunization where the frequencies of antigen specific cells are low (and in Suppl figures).

Response: We did not show any control tetramer staining in the original version of our manuscript as the use of pMHCII tetramers is now well documented, however when the use of these tetramers was established in the laboratory we compared our tetramer of interest with the control pMHCII tetramers for I-A(b) human CLIP 87-101 PVSKMRMATPLLMQA. We indeed made a mistake for not presenting these data in the first place thus they are depicted in Figure 8 for Reviewers (Fig R8). Fig R8 shows that no hCLIP 87-101 IAb tetramer⁺ cells can be found in 1W1K-OVA immunized mice both in draining LN and in the spleen 30 days after immunization. In addition, almost no positive cells using 1W1K-IAb tetramer are observed in LN and spleen from unimmunized and day 30 OVA-immunized animals (Fig R8).

Action: We have added a supplementary figure (Fig S1) using the data presented in Fig R8 to clearly show that we track 1W1K-specific Th cells and have modified the text accordingly page 6.

4. The identification of memory B cells is not sufficient. The more standard approach is to have a dump channel using anti-Gr1 to remove antibody-antigen binding myeloid cells that are especially as issue in the spleen. Moreover CD38 is used to separate antigen-specific memory from GC B cells that are not assessed. Similarly as outlined above the authors should show some non-immunized mice as background staining, this is most pertinent to determine if the NP-low binding cells in fig 5a spleen are indeed antigen specific.

Response: We think this reviewer comment is fair and apologize that we were not clear in the original version of our manuscript. As stated before, regarding the first comment of Reviewer#1, we indeed did not present any control staining in the original version of the manuscript as the use of labelled NP to detect NP-specific B cells is well documented. However, as depicted in Fig R1, we have established the use of NP-PE in unimmunized mice or in mice immunized with an irrelevant protein (OVA) emulsified in IFA/CpG in which almost no NP-specific B cells can be detected in draining LN and in spleen. Moreover, we also apologize for not showing in the first place the phenotype of the NP⁺ B cells, especially GL-7 and CD38 expression. Again, as stated above, this was a mistake and Fig R5 shows that at day 30 post-immunization, mainly all draining LN and all spleen NP⁺ B cells are GL-7^{neg}CD38⁺ cells, thus B cells that are not GC B cells. Finally, based on this Reviewer comment and the first comment of Reviewer#1, we realized that our gating strategy to total B cells in order to select NP-specific B cells by looking at IgD⁺NP⁺ cells was not clear. In figure 9 for reviewers (fig R9), the gating strategy is presented. In order to clearly identify B cells, our strategy is to dump out dead cells, cell doublets and CD4⁺/CD8⁺ cells. In a second step, we positively gate on total B cells by gating on CD138/B220⁺ cells allowing the exclusion of non specific cells (Fig R9).

Action: We have modified Figure 5 of the revised manuscript using the data presented in Fig R1 to show that the NP-PE gate is accurate and have modified the text accordingly page 11 and page 35. Second, we have added a supplementary figure (Fig S8) using the data presented in Fig R5 to show that we track NP⁺ memory B cells both in the draining LN and in the spleen and have modified the text accordingly page 11. Finally, we have added a supplementary figure (Fig S7) using the data presented in Fig R9 to show the gating strategy

used to positively gate on total B cells and have modified the text accordingly page 11 and 35.

5. Figure 6. The differences in cytokine production between the restimulated dLN and spleen populations seems very marginal at best as depicted from panels b and c. The fold change shown in panel a does not appear to be an accurate representation of the small differences and should be removed. The cytokine production should be quantified using an independent method such as real time PCR. Even if the small differences in cytokine production are real, the data in fig 7 shows that the lower cytokine producing dLN mTfh cells are better at supporting class switching, a finding that raises doubt on any interpretation of fig 6.

Response: We have to confess that this Reviewer comment is really not clear to us. This Reviewer seems concerned about the data presented Figure 6 and appears to have some difficulty appreciating the way the experiments were presented. Indeed, it is asked by this Reviewer to quantify cytokine production with an independent method such as real time PCR, which is actually what we performed and what was presented in panel a Figure 6. Moreover, what also seems a concern for this Reviewer is that the differences of cytokine production observed between draining LN and spleen cells are not to the same extent between panel a and panel c. We agree with this observation but also want to emphasize on the fact that the experimental procedures were not similar in different ways: in vitro stimulation (anti CD3/CD28 vs PMA/ionomycin), time of stimulation (48h vs 4h), detected products and detection method (mRNA transcripts by real time PCR vs protein by Flow cytometry). Overall, we think that this could explain why the magnitude of differences between draining LN and spleen cells was different between panel a and panel c. Anyhow, the difference observed using the two experimental procedures was not divergent: using two different experimental approaches, we found that restimulated spleen cells contained more mRNA IL-21 and IFN- γ transcripts, which correlated with more of these protein cytokines after PMA/ionomycin stimulation. Thus, overall, we think that our conclusion, which is 'upon restimulation, circulating mTfh have greater capacities to produce cytokines than the local mTfh', still stands.

6. Following on from point #5, the mechanism by which the local and circulating mTfh cells would induce GC in the spleen and IgG PCs in the dLN is unclear and a major limitation of the paper. The authors propose a role for IL21 in promoting the GCs, yet this disregards the very potent function of IL21 inducing PC differentiation and the lack of extrafollicular plasmablasts in the IL21 KO mice after immunization. Moreover the function of IL21 in GCs is only apparent late after immunization, not at the early stages examined here (ref 9, 10)

Response: The intrinsic role of IL-21 in B cells was demonstrated in ref 9 and ref 10. The authors were able to demonstrate that 'IL-21 acts in a B cell intrinsic fashion to control GC B cell formation' (ref 9) and that 'IL-21 regulates GC B cell differentiation and proliferation' (ref 10). In addition, it was shown in reference 10 that IL-21KO mice had a decrease in the early extrafollicular antibody-secreting cells (ASC) as compared to wild-type mice, but the authors also demonstrated that IL-21KO mice showed a decrease high-affinity GC-derived ASC response due to GC abnormality. Thus, like authors from ref 9 and 10, we think that the main role of IL-21, at least in T-dependent B cell response, is to promote GC B cell differentiation. Moreover, as stated above in response to one comment of Reviewer#1, we made the choice to monitor the NP⁺ B cell response early after immunization. It is true that in ref 9 and 10, the authors monitored the B cell response at later timepoint but it was in primary response. Since our idea was to be the closest to the physiology and since we studied the impact of memory cells, we therefore decided to assess NP-specific B cell response 5 days post-immunization. Again, this choice was made based on the literature that shows that the peak of the secondary NP-specific B cell response is reached as of day 4 post-boost (for example see McHeyzer-Williams *et al.*, J Exp Med, 2000).

7. Model. The overall conclusion is that mTfh cells are held in the local dLN by cognate peptide-MHCII interactions with memory B cells. For this to be the case the antigen specific Tfh and memory B cells (presumably affinity matured) need to perceive this interaction as memory sustaining, as opposed to an immune challenge resulting in activation and effector outcomes such as PC differentiation. The authors do not provide a convincing model to explain how the cells differentially control these outcomes, given that this is a antigen-specific interaction.

Response: We went back to the original version of our manuscript and thank the reviewer for this comment since we realized that we were initially not clear. We are not the first to show that cognate interaction can occur even in the memory phase and that this cognate interaction results in memory maintenance in contrast to activation of the engaged cells. To us, the best example remains the tissue-resident memory CD8⁺ T cells (TRM) in non-lymphoid organs. One striking feature of these cells is expression of CD69, as a marker of cognate interaction, which eventually participates to their retention (Mackay LK *et al.*, JI, 2015). Regarding our studies, we found that local mTfh do express CD69 even day 120 after immunization and to a level that was closely similar to the one of mTfh 30 days post-immunization (Fig S4). Overall, we thank the reviewer for pointing this out and have modified the text of the revised version of our manuscript by clarifying the proposed model in pages 8/17/18 and 30.

8. The experiments and figures are complex (transfer, TCR Tg, different routes of challenge etc) making some aspects difficult to read and compare. To help the reader the authors should consider providing some schema for the experimental flow on the figures associated with some of the more complex experiments.

Response: It is totally true that some of the experimental procedures used are complex. In order to simplify the reading of the revised version of the manuscript Figure 10 for reviewers (Fig R10) and Figure 11 for reviewers (Fig R11) are presented and have been added as supplementary figures (Fig S10 and Fig S11). The text of the revised manuscript has been modified accordingly pages 13/14/15/36/37.

9. The data in Fig 3 show on the whole that the Vb gene usage is very similar between the antigen-specific local and circulating Tfh cells. However the RT-PCR in panel d (showing very similar Vb14 %) does not match the FACS estimation in panel e showing a pronounced difference between spleen and dLN. One of these measurements must be inaccurate.

Response: Regarding the difference observed at the mRNA vs protein level for V β 14 usage, we think this comment is inadequate. Indeed, V β 14 usage in the dLN is similar between Fig 3D and Fig 3E. The only difference corresponds to the usage from spleen cells. We believe that this difference can be explained by the fact that RT-qPCR were performed on materials from 10 pooled different mice while flow cytometry data were performed on single animal as described in the methods section and in the figure legend. Anyhow, the conclusion of this 2 panels is that the V β 14 gene segments is dominant in 1W1K-specific memory Th cells in the dLN and in the spleen, which can not be argued and still stands.

10. It is mentioned a couple of times in the text, that the memory B cells and high affinity, but this is not actually shown, and only NP15 (total NP) is measured in the ELISAs.

Response: It is true that we monitored and presented only the serum NP-specific IgG in the original version for NP15. However, we also monitored the serum NP25-specific IgG. As presented in Figure 12 for Reviewers (Fig R12), we depicted the relative NP15/NP25 ratio that shows that local mTfh indeed promote more high affinity NP-specific IgG than circulating mTfh.

Action: Fig R12 can be added to the main figure or as a supplementary figure if deemed necessary.

Figure 1 for Reviewers

Gating of NP-specific B cells in dLN and in spleen using NP-PE

C57BL/6 mice were immunized sc with 100 μ g OVA or NP-OVA in IFA+CpG.

30 days after, dLN and spleen from immunized mice or from naive mice were collected.

NP and IgD staining on $CD4^-CD8^-B220/CD138^+$ cells is presented.

Frequency denotes the mean value of groups and SEM. (n=5/group)

Figure 2 for Reviewers

CD80 expression at the surface of NP⁺ memory B cells

C57BL/6 mice were immunized sc with 100 μ g NP-OVA in IFA+CpG. 30 days after, dLN and spleen from immunized mice were collected. CD80 expressions was monitored at the surface of CD4⁻CD8⁻B220/CD138⁺ NP⁺ cells. CD80 gMFI is depicted in a.

In b, expression of isotype control and CD80 is presented as well as the frequency of CD80⁺ cells.

P<0.01; *P<0.001

Figure 3 for Reviewers

PD-L2 expression at the surface of NP⁺CD8⁻ memory B cells

C57BL/6 mice were immunized sc with 100 μ g NP-OVA in IFA+CpG. 30 days after, dLN and spleen from immunized mice were collected. PD-L2 expression was monitored at the surface of CD4⁻CD8⁻B220⁺/CD138⁺NP⁺CD8⁻ cells.

Figure 4 for Reviewers

Dead cells after in vitro stimulation

C57BL/6 mice were transferred iv with purified CD4⁺CD45.1⁺ naïve OT-II cells and sc immunized 24h later with OVA in IFA/CpG. 30 days after, dLN and spleen CD4⁺CD45.1⁺CD44⁺CXCR5⁺ OT-II cells were purified and stimulated in vitro with PMA/ionomycin. Viability of activated cells was monitored using viability dye.

Figure 5 for Reviewers
Kinetics of NP⁺ B cells

C57BL/6 mice were immunized sc with 100 μ g NP-OVA in IFA+CpG. GL-7 and CD38 expression were monitored at the surface of NP⁺ B cells. In a is presented the expression level of dLN and spleen cells day 14 and 30 post-immunization. In b are presented the kinetics of NP-specific IgD⁻CD38⁻GL-7⁺ B cells after immunization (mean \pm SEM, n \geq 5/time point)

Figure 6 for Reviewers

Kinetics of GC reaction in dLN after 1W1K-OVA immunization

C57BL/6 mice were immunized sc with 1W1K-OVA in IFA+CpG. 14 and 30 days after, dLN from immunized mice or from naive mice were collected and confocal microscopy studies were performed using anti-IgD (blue), anti-GL-7 (green) and CD4 (red) mAb. LN were harvested into PLP buffer (0.05M phosphate buffer containing 0.2ml-lysine [pH 7.4], 2 mg/ml NaIO₄, 10 mg/ml paraformaldehyde), fixed overnight and dehydrated in 30% sucrose prior to embedding in OCT freezing media (Sakura Finetek). Frozen sections were cut on a CM1950 Cryostat. Sections were stained in PBS/1% BSA. Images were acquired on a Apotome ZEISS Inv. (scale bars; 200 μ m). Number of GCs per mm² in LN sections is depicted. n=8/conditions. mean+SEM. ns, non significant; **P<0.01

Figure 7 for Reviewers

Ki-67 expression is similar at day 30 or day 90 post-immunization in mTfh cells

30 days after sc immunization, dLN and spleen were analyzed for intracellular expression of Ki-67 among 1W1K-specific Tfh cells at day 30 and day 90 post-immunization. ns, non-significant

CD19⁻CD8⁻CD4⁺

Figure 8 for Reviewers

Gating of 1W1K-specific Th cells in dLN and in spleen using 1W1K-I-Ab tetramer

C57BL/6 mice were immunized sc with 100 μ g OVA or 1W1K-OVA in IFA+CpG.

30 days after, dLN and spleen from immunized mice or from naive mice were collected and were analyzed for the detection of 1W1K-IAb⁺CD44⁺ and hCLIP87-101-IAb⁺CD44⁺ Th cells.

Frequency denotes the mean value of groups and SEM (n=5/group).

Figure 9 for Reviewers

Gating strategy to total B cells in dLN and in spleen 30 post-immunization

C57BL/6 mice were immunized sc with 100 μ g NP-OVA in IFA+CpG.

30 days after, dLN and spleen from immunized mice were collected and were analyzed for

the detection of total B cells by excluding dead cells, CD4/CD8⁺ cells and

by positively selecting C138/B220⁺ cells.

Figure 10 for Reviewers

Experimental scheme of *in vivo* transfer and immunization related to Figure 7

sc, subcutaneous; iv, intravenously; ip, intraperitoneally.

Figure 11 for Reviewers
Experimental schemes related to Figure 8

Figure 12 for Reviewers

NP15- NP25-specific IgG response

10,000 dLN or spleen CD4⁺CD45.1⁺CD44⁺CXCR5⁺ OT-II cells from day 30 sc immunized mice were transferred into mice previously immunized 30 days before with NP-KLH in SAS either ip or sc (see figure S9 for experimental scheme). Then, mice were ip immunized with NP-OVA. 5 days after, sera of mice were collected to perform ELISA. Relative ratio of NP15 and NP25 is depicted. Each dot represents an individual mouse; horizontal lines denote the mean value of groups and SEM

REVIEWERS' COMMENTS:

Reviewer #1 (Remarks to the Author):

The authors nicely addressed my critiques. I do not have any other concerns and support the publication in Nature Communications.

Reviewer #2 (Remarks to the Author):

Overall I feel that the author have adequately addressed my concerns and i am in favor of publication as is.

Response to Reviewers

Comments:

Reviewer #1 (Remarks to the Author):

The authors nicely addressed my critiques. I do not have any other concerns and support the publication in Nature Communications.

Reviewer #2 (Remarks to the Author):

Overall I feel that the author have adequately addressed my concerns and i am in favor of publication as is.

Response: We would like to thank the reviewers for his/her comments that clearly state that we have addressed all issues raised by them.